# Tonic ubiquitination of the central body weight regulator melanocortin receptor 4 (MC4R) promotes its constitutive exit from cilia

**Irene Ojeda-Naharros[1,2], Tirthasree Das[1,2], Ralph A. Castro[1,2], J. Fernando Bazan[3,4], Christian Vaisse[5], Maxence V. Nachury[1,2]***

1 Department of Ophthalmology, University of California San Francisco, California, United States of America, 2 Cardiovascular Research Institute, University of California San Francisco, California, United States of America, 3 Unit for Structural Biology, VIB-UGent Center for Inflammation Research, Ghent, Belgium, 4 ℏ bioconsulting llc, Stillwater, Minnesota, United States of America, 5 Diabetes Center, University of California San Francisco; San Francisco, California, United States of America

* maxence.nachury@ucsf.edu

**Data Availability Statement:** All relevant data are within the paper and its Supporting Information files.

## Abstract

The G protein-coupled receptor (GPCR) melanocortin receptor 4 (MC4R) is an essential regulator of body weight homeostasis. MC4R is unusual among GPCRs in that its activity is regulated by 2 opposing physiological ligands, the agonist α-MSH and the antagonist/inverse agonist AgRP. Paradoxically, while MC4R localizes and functions at the cilium of hypothalamic neurons, the ciliary levels of MC4R are very low under unrestricted feeding conditions. Here, we find that the constitutive activity of MC4R is responsible for the continuous depletion of MC4R from cilia and that inhibition of MC4R's activity via AgRP leads to a robust accumulation of MC4R in cilia. Ciliary targeting of MC4R is mediated by its partner MRAP2 and the constitutive exit of MC4R from cilia relies on the sensor of activation β-arrestin, on ubiquitination, and on the BBSome ciliary trafficking complex. Thus, while MC4R exits cilia via conventional mechanisms, it only accumulates in cilia when its activity is suppressed by AgRP.

## Introduction

The leptin-melanocortin axis relays the levels of adipose reserves to the central nervous system to maintain body weight at a set value [1,2]. Leptin is secreted by adipose tissues and primarily sensed in the hypothalamus, where it promotes the release of α-melanocyte stimulating hormone (α-MSH) and inhibits the secretion of agouti-related peptide (AgRP) [3]. The neuropeptides α-MSH and AgRP are subsequently detected by the G protein-coupled receptor (GPCR) melanocortin receptor 4 (MC4R), chiefly in the paraventricular nucleus (PVN) of the hypothalamus. The molecular tuning of MC4R's activity by its agonist α-MSH and its inverse agonist AgRP adjusts the calorie intake to match the long-term energy needs of the organism.

**Funding:** This work was supported by the National Institutes of Health (GM089933 to MVN), Vision Core Grant (EY031462 to MVN), the American Diabetes Association (1-20-VSN-03 to MVN), National Institutes of Health (R01DK060540 and R01DK106404 to CV), Vision Core Grant (EY002162 to MVN), Research to Prevent Blindness (Unrestricted Grant to UCSF), the Swiss National Science Foundation (P400PB_191097 to ION), the UCSF Program for Breakthrough Biomedical Research (7000/7002124 to ION) and HDFCCC core grant (P30CA082103 to MNV). The funders had no role in study design, data collection and analysis, decision to publish, or preparation of the manuscript.

**Competing interests:** The authors have declared that no competing interests exist

**Abbreviations:** AgRP, agouti-related peptide; α-MSH, α-melanocyte stimulating hormone; BBS, Bardet–Biedl syndrome; GPCR, G protein-coupled receptor; MC4R, melanocortin receptor 4; MRAP2, melanocortin receptor accessory protein 2; PBS, phosphate-buffered saline; PVN, paraventricular nucleus; SSTR3, somatostatin receptor 3; TM, transmembrane; UBD, ubiquitin-binding domain.

Primary cilia are specialized signaling antennas that project from the cell body to detect and organize diverse signaling pathways [4–6]. Malfunctioning cilia result in syndromic obesity in several ciliopathies [7,8] and genetic studies in mice have shown that ablation of neuronal primary cilia in adult brain is sufficient to disrupt body weight homeostasis [9,10]. The signaling pathway organized by neuronal cilia for body weight homeostasis remained largely elusive until the discovery that MC4R is localized at primary cilia of hypothalamic neurons in mice and rats [11,12]. We previously demonstrated that MC4R activity at the primary cilia of PVN neurons is required for its ability to regulate body weight [13].

Enrichment of MC4R in cilia requires its interaction partner, the single pass membrane protein melanocortin receptor accessory protein 2 (MRAP2) [13] and deletion of MRAP2 or mutations that block ciliary targeting of MC4R result in profound obesity in human patients and in mice [13–20]. A current paradox is that, even though MC4R needs to target to cilia in order to regulate feeding behavior, ciliary levels of MC4R are low in the adult brain under ad lib fed conditions [11,12,21].

The primary cilium concentrates a number of signaling molecules at its membrane and the enrichment of signaling molecules inside cilia is often dynamically gated by pathway activation [22–24]. For instance, in Hedgehog signaling, the receptor Patched 1 undergoes exit from cilia once bound to its ligand, while the GPCR Smoothened (SMO) accumulates inside cilia once the pathway becomes activated [25]. Regulation of the exit rate appears to be the predominant mechanism regulating the dynamic accumulation of signaling factors inside cilia [5]. Signaling receptors that need to be removed from cilia become marked by ubiquitin, a small polypeptide with roles in protein degradation and regulation [26–28]. For the prototypical ciliary GPCR somatostatin receptor 3 (SSTR3) and the Hedgehog-responsive GPCR GPR161, the conformational sensor β-arrestin 2 directs the ubiquitination machinery to activated receptors. Ubiquitinated signaling receptors are then recognized and ferried out of cilia by the BBSome, a complex of proteins mutated in the obesity disorder Bardet–Biedl syndrome (BBS), with the aid of the ubiquitin reader TOM1L2 [29–31]. The functional relationship between MC4R and BBS remains to be determined.

In this study, we sought to determine how ciliary MC4R levels are regulated. By recapitulating MC4R trafficking in a cell culture system devoid of neuronal connections, we studied the cilia- and MC4R-intrinsic regulation of its trafficking. While MC4R exit from cilia depends on its ubiquitination and on the BBSome, we made the surprising finding that the high tonic activity of MC4R results in its constitutive exit from cilia and that MC4R levels drastically increase upon inhibition of the tonic activity by inverse agonists. Thus, in diametrically opposite fashion to SMO, the ciliary levels of MC4R are anticorrelated with its activity.

## Results

### The constitutive activity of MC4R is responsible for its low ciliary abundance

To study the ciliary dynamics of MC4R, we generated a stable isogenic cell line in mouse kidney IMCD3 cells expressing low and stoichiometric levels of MRAP2-3xFLAG (MRAP2$^{3FLAG}$) and MC4R-3xmNeonGreen (MC4R$^{3NG}$). MRAP2 and MC4R were expressed from the same mRNA, separated by a T2A self-cleaving peptide, and transcript expression was driven by the attenuated EF1α promoter (pEF1α$^{\Delta}$), which we previously used to recapitulate the endogenous expression levels of the neuronal ciliary GPCR SSTR3 [32]. The MRAP2/MC4R cell line displayed ciliary MC4R localization at variable and barely detectable levels (**Figs 1A, 1B, and S1A**), recapitulating observations in rodents [11,12]. When compared to a cell line that expresses GPR161$^{3NG}$ from a considerably weaker promoter than pEF1α$^{\Delta}$, the mRNA levels of

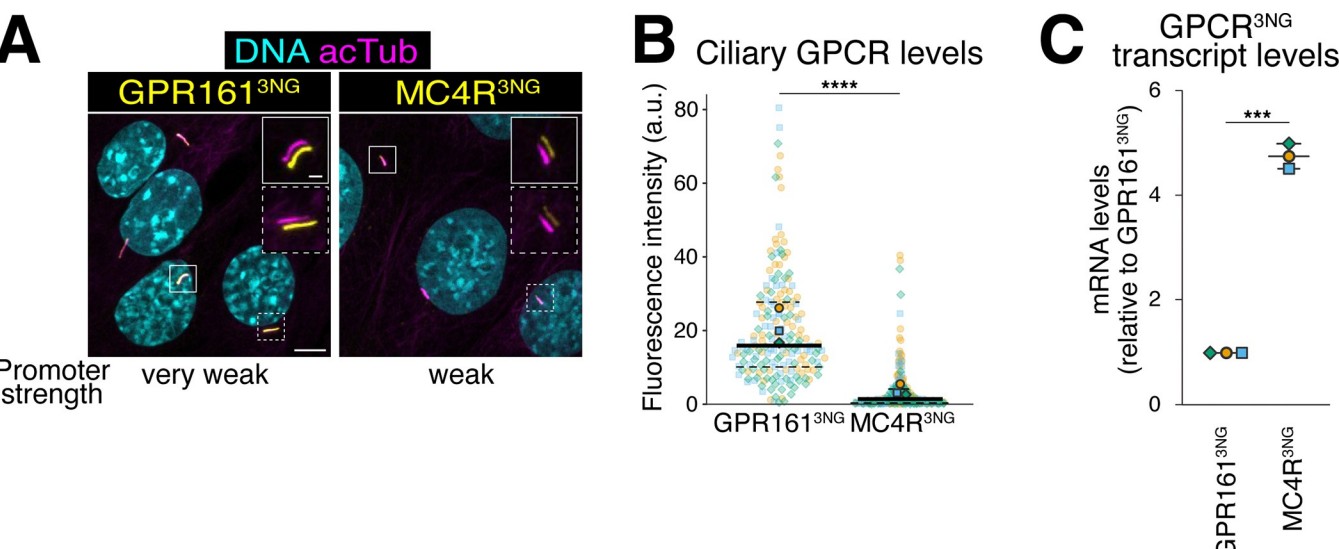

**Fig 1. The ciliary abundance of MC4R is considerably lower than that of other GPCRs.** (**A**) Representative images of clonal IMCD3 lines stably expressing GPR161[3NG] under the control of the ultra-weak δ-crystallin promoter (pCrysδ) or MRAP2[3FLAG]/MC4R[3NG] under the control of the attenuated promoter pEF1α[A]. Serum-starved cells were fixed and stained for acetylated tubulin (acTub, magenta) and DNA (cyan). GPR161[3NG] and MC4R[3NG] were visualized through the intrinsic fluorescence of NG (yellow). The yellow and magenta channels are shifted to facilitate visualization of ciliary signals in the insets. Scale bars: 5 μm (main panel) and 1 μm (inset). (**B**) Superplots of the ciliary fluorescence intensities of GPR161[3NG] vs. MC4R[3NG]. $n = 3$ independent experiments. Data points belonging to each different experiment are encoded by translucent points of different color and shape. The average of each experiment is represented by solid points. A solid line has been used to represent the median of all combined data and dashed lines to represent the interquartile values. Asterisks indicate statistical significance values calculated by a parametric unpaired $t$ test on individual cilia (**** $p < 0.0001$). All underlying data are found S1 Data. (**C**) Relative transcript levels of *GPR161[3NG]* vs. *MC4R[3NG]* assessed by qRT-PCR amplification of 3NG. *GAPDH* was used to normalize signals between samples. Values are displayed relative to the mRNA levels in the IMCD3-[GPR161[3NG]] cell line. $n = 3$ independent biological replicates. Each point represents the mean of 3 averaged qRT-PCR runs of a biological replicate. Error bars represent standard deviations. Asterisks indicate statistical significance values calculated by a parametric ratio-paired $t$ test (*** $p < 0.001$). All underlying data are found S1 Data. GPCR, G protein-coupled receptor; MC4R, melanocortin receptor 4.

MC4R[3NG] were 5-fold higher than those of GPR161[3NG], yet the ciliary intensity of MC4R[3NG] was over 5-fold lower than that of GPR161[3NG] (**Figs 1B, 1C, and S1A**). Leveraging prior estimates that 1,200 molecules of GPR161[3NG] reside in each cilium on average [32], we estimate 220 MC4R[3NG] molecules per cilium on average. The striking difference in ciliary levels between MC4R[3NG] and GPR161[3NG] indicates either that MC4R enters cilia very inefficiently or that MC4R continuously enters and exit cilia at equivalent rates. While counterintuitive, this latter scenario is well exemplified by SMO in the absence of Hedgehog ligands [5].

GPCRs undergo regulated exit from cilia when exposed to ligands [5]. However, MC4R ligands are absent in our cell culture system, as all experiments are carried out in the absence of serum and kidney epithelial cells do not secrete MC4R ligands. Noticeably, MC4R possesses a high tonic activity, possibly because its extracellular N-terminal tail acts as a tethered partial agonist [33–35]. We thus hypothesized that tonically active MC4R may undergo constitutive exit from the cilium in the absence of ligands. To test this hypothesis, we suppressed the tonic activity of MC4R with the natural inverse agonist AgRP [36] or the synthetic inverse agonist HS014 [37]. We observed that both inverse agonists promoted a steady increase of the ciliary levels of both MRAP2 and MC4R until 8 h of treatment, after which the accumulation progressed at a slower rate (**Fig 2A**). Immunoblotting revealed that the total amounts of MRAP2 and MC4R increased after treating cells with inverse agonists (**Fig 2B and 2C**), presumably because the tonic activity of MC4R leads to some level of constitutive degradation. However, we note that while the addition of AgRP ultimately increased ciliary levels of MC4R and MRAP2 by 11- and 25-fold, the total protein levels of MC4R and MRAP2 increased at most by

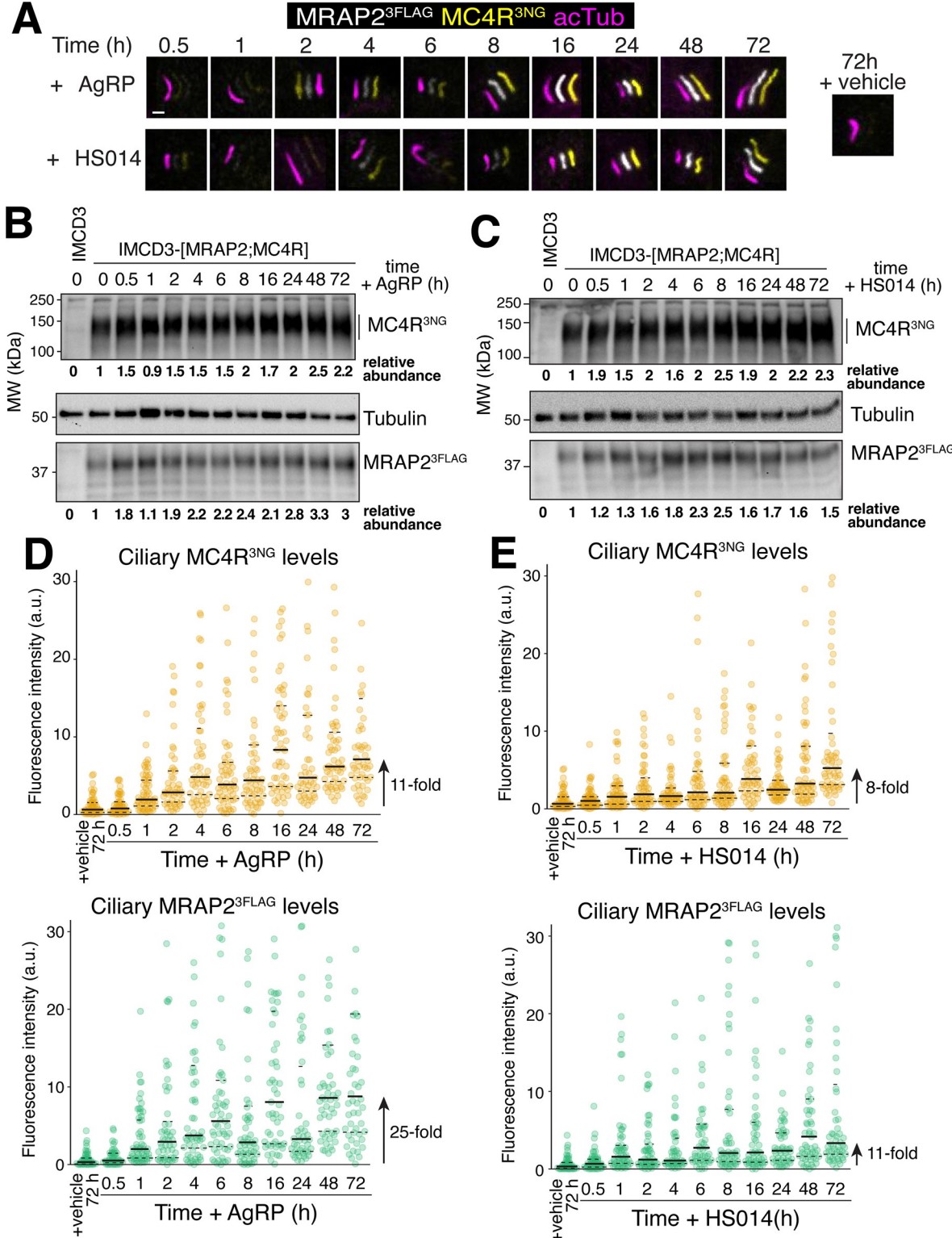

**Fig 2. Blocking the tonic activity of MC4R increases its ciliary abundance.** (**A**) Representative images of IMCD3-[MRAP2^3FLAG/MC4R^3NG] cells treated with either HS014, AgRP or vehicle were fixed at the indicated time points. Cells were stained for acetylated tubulin (acTub, magenta) and FLAG (MRAP2^3FLAG, white). MC4R^3NG was visualized through the intrinsic fluorescence of NG (yellow). The white, yellow, and magenta channels are shifted to facilitate visualization of ciliary signals. Scale bar: 1 μm. (**B**) IMCD3-[MRAP2^3FLAG/MC4R^3NG] cells treated with AgRP or vehicle were lysed at the indicated time points. Cell lysates were resolved by SDS-PAGE and probed for NG

(MC4R[3NG]), Tubulin, and FLAG (MRAP2[3FLAG]) to assess total protein levels. MC4R[3NG] and MRAP2[3FLAG] band intensities were measured and normalized to the tubulin density (loading control). The abundances of MC4R[3NG] and MRAP2[3FLAG] relative to their levels at t = 0 are indicated below the blots. (**C**) IMCD3-[MRAP2[3FLAG]/MC4R[3NG]] cells treated with HS014 or vehicle were lysed at the indicated time points. Cell lysates were resolved by SDS-PAGE and probed for NG (MC4R[3NG]), Tubulin, and FLAG (MRAP2[3FLAG]) to assess total protein levels. MC4R[3NG] and MRAP2[3FLAG] band densities were measured and normalized to the tubulin density (loading control). The abundances of MC4R[3NG] and MRAP2[3FLAG] relative to their levels at t = 0 are indicated below the blots. (**D**) Plots of ciliary MC4R[3NG] (orange) and MRAP2[3FLAG] (green) in MRAP2[3FLAG]/MC4R[3NG] cells treated for the indicated time points with either AgRP or vehicle. All underlying data are found S1 Data. (**E**) Plots of ciliary MC4R[3NG] (orange) and MRAP2[3FLAG] (green) in MRAP2[3FLAG]/MC4R[3NG] cells treated for the indicated time points with either HS014 or vehicle. All underlying data are found S1 Data. AgRP, agouti-related peptide; MC4R, melanocortin receptor 4; MRAP2, melanocortin receptor accessory protein 2.

only 2.5- and 3.3-fold (**Fig 2D**). Similarly, HS014 addition increased the ciliary levels of MC4R and MRAP2 by 8- and 11-fold while increasing total protein levels by at most 2.5-fold (**Fig 2E**). The much greater effect size of inverse agonists on ciliary abundance than on total protein levels suggests that inverse agonists strongly and specifically suppress the ciliary exit of MRAP2/MC4R, while modestly reducing degradation of MRAP2/MC4R.

To mimic the in vivo situation where MC4R is concurrently exposed to 2 opposing physiological ligands, we stimulated MC4R with increasing concentrations of its endogenous agonist α-MSH in the presence of a constant concentration of AgRP. While increasing concentrations of α-MSH did reduce the ciliary levels of MC4R and MRAP2, even a 20-fold molar excess of α-MSH over AgRP was not sufficient to reduce ciliary MC4R levels to the baseline levels in the absence of ligands (**Figs 3 and S2**). Thus, besides functioning as an inverse agonist of MC4R that reduces the constitutive activity of MC4R and elevates the ciliary levels of MC4R in the absence of other ligands, AgRP can significantly elevate the ciliary levels of MC4R in the presence of any physiological concentration of α-MSH owing to its potent antagonist activity [38,39].

## MRAP2 facilitates the import of MC4R into cilia

As MRAP2 is required for the enrichment of MC4R in cilia [13] and MRAP2 overexpression decreases the tonic activity of MC4R [40], we sought to determine whether MRAP2 may promote the ciliary enrichment of MC4R by inhibiting its tonic activity. We first leveraged in silico structural modeling with AlphaFold2-multimer (as implemented on ColabFold; [41–43]) to determine how MRAP2 may block the tonic activity of MC4R. This coevolution-reliant deep-learning algorithm predicted with very high confidence the hydrophobic contacts between the transmembrane (TM) segment of MRAP2 and the TM5 and TM6 helices of the 7-TM bundle of MC4R (**Fig 4A**). This prediction was fully corroborated by the improved AlphaFold3 program [44]. We note that the membrane topology of MRAP2 in the MRAP2/MC4R complex is robustly N-out/C-in and this orientation is retained in dimers of MRAP2 (**S1B Fig**). These findings stand in contrast with the proposed antiparallel packing of MRAP2 dimers based on overexpression studies [45,46]. Congruent with the structural models, probing the orientation of MRAP2 in the IMCD3-[MRAP2;MC4R] cell line unequivocally showed that MRAP2 adopts an N-out/C-in topology when expressed at low levels with a minimal tag (**Fig 4B**).

A surprising finding of the MRAP2/MC4R model is that a short segment from the MRAP2 cytoplasmic tail (E84-G97, **Fig 4A**) appears to insert into the site occupied by Gα in activated GPCRs, thus suggesting a possible basis for the inhibition of MC4R tonic activity by MRAP2. In further support of this hypothesis, the same segment of the MRAP2 C-tail also occludes the Gα activating pocket in the model of MRAP2/MC4R in complex with the agonist α-MSH (**S1C Fig**) but not in the model of MRAP2/MC4R in complex with AgRP (**Fig 4C**). However, deleting the MC4R blocking motif from MRAP2 C-tail (MRAP2[Δblock]) did not affect the ability

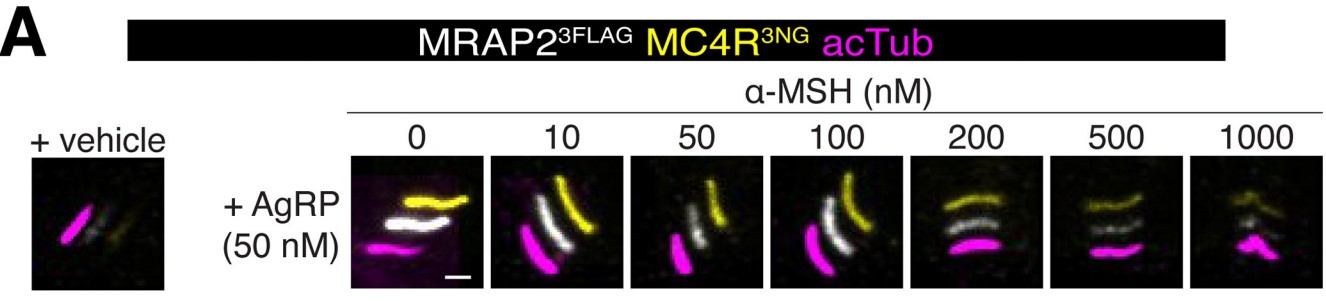

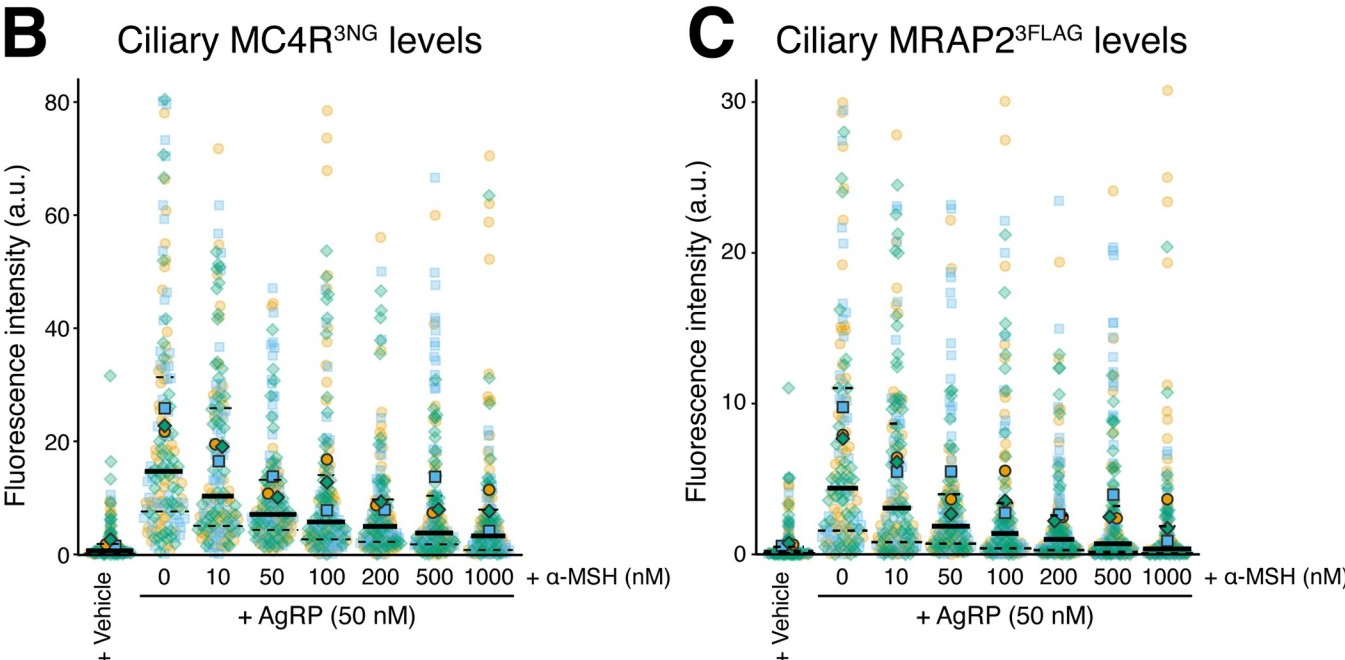

**Fig 3. Effects of co-treatment with AgRP and α-MSH on the ciliary levels of MC4R and MRAP2.** (**A**) Representative images of IMCD3-[MRAP2$^{3FLAG}$/MC4R$^{3NG}$] cells treated with either vehicle or 50 nM AgRP and increasing concentrations of α-MSH (from 0 nM to 1,000 nM). Cells were stained for acetylated tubulin (acTub, magenta) and FLAG (MRAP2$^{3FLAG}$, white). MC4R$^{3NG}$ was visualized through the intrinsic fluorescence of NG (yellow). The white, yellow, and magenta channels are shifted to facilitate visualization of ciliary signals. Scale bar: 1 μm. (**B**) Superplot comparing the ciliary fluorescence intensity of MC4R$^{3NG}$ in IMCD3-[MRAP2$^{3FLAG}$;MC4R$^{3NG}$] treated with either vehicle or 50 nM AgRP ± α-MSH at different concentrations. $n$ = 3 independent experiments. Data points belonging to each different experiment are encoded by translucent points of different color and shape. The average of each experiment is represented by solid points. A solid line has been used to represent the median of global data and dashed lines to represent the interquartile values. All underlying data are found S1 Data. (**C**) Superplot comparing the ciliary fluorescence intensity of MRAP2$^{3FLAG}$ in IMCD3-[MRAP2$^{3FLAG}$;MC4R$^{3NG}$] treated with either vehicle or 50 nM AgRP ± α-MSH at different concentrations. $n$ = 3 independent experiments. Data points belonging to each different experiment are encoded by translucent points of different color and shape. The average of each experiment is represented by solid points. A solid line has been used to represent the median of global data and dashed lines to represent the interquartile values. All underlying data are found S1 Data. AgRP, agouti-related peptide; MC4R, melanocortin receptor 4; MRAP2, melanocortin receptor accessory protein 2.

of MRAP2 to increase the ciliary levels of MC4R (**Figs 4D, 4E, and S1D**). Thus, the putative MC4R blocking motif of MRAP2 does not appear to influence the constitutive exit of MC4R from cilia.

We note that the unstructured N-terminus of MRAP2 is predicted to make extensive and distinct contacts with both AgRP (**Fig 4C**) and α-MSH (**S1C Fig**), reminiscent of the contacts made by MRAP1 with ACTH in the structure of the MRAP1/MC2R/ACTH complex [47]. The

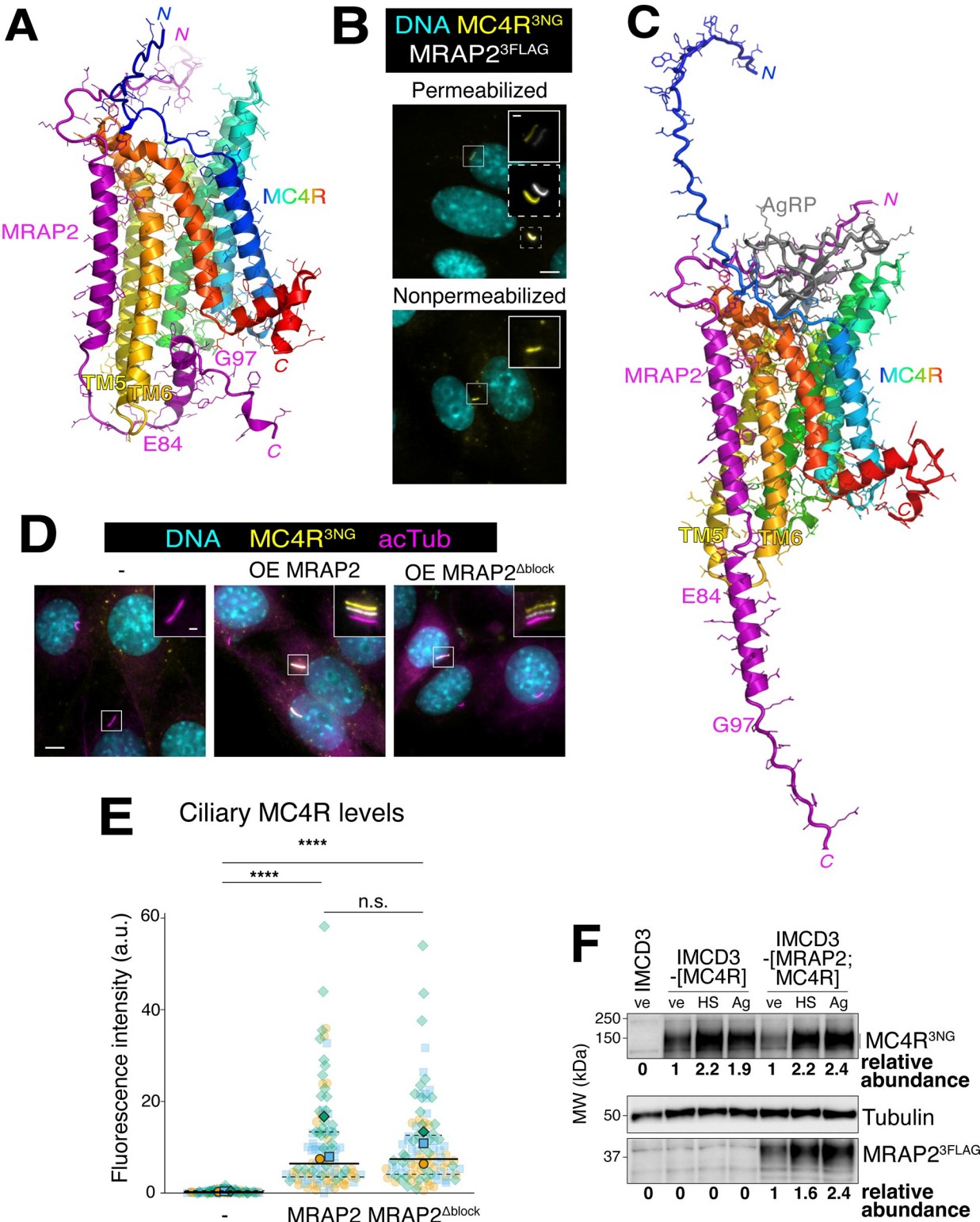

**Fig 4. MRAP2 does not suppress MC4R exit by blocking its tonic activity. (A)** AlphaFold2.3 model of human MC4R/MRAP2. The predicted MC4R blocking motif extends from E84 to G97. (**B**) Representative images of IMCD3-[MRAP2³FLAG;MC4R³NG] cells stained for FLAG in permeabilized vs. nonpermeabilized conditions, (MRAP2³FLAG, white) and DNA (cyan). MC4R³NG was visualized through the intrinsic fluorescence of NG (yellow) and provides a reference for primary cilium. The white and yellow channels are shifted to facilitate visualization of ciliary signals in the insets. Scale bars: 5 μm (main panel) and 1 μm (inset). (**C**) AlphaFold2.3 model of human MC4R/MRAP2 in a 1:1:1 complex

with human AgRP. (**D**) Representative images of IMCD3-[MC4R^3NG] untransfected or transiently transfected cells with MRAP2^3FLAG or the MRAP2 version lacking the candidate MC4R inhibitory motif MRAP2^Δblock-3FLAG. DNA (cyan), MRAP2^3FLAG or MRAP2^Δblock-3FLAG (white), MC4R^3NG (yellow), cilia (acetylated tubulin, magenta). The white, yellow, and magenta channels have been shifted in the insets for better visualization. Scale bars: 5 μm (main panel) and 1 μm (inset). (**E**) Superplots of the ciliary fluorescence intensities of MC4R^3NG in IMCD3-[MC4R^3NG] untransfected or transiently transfected cells with MRAP2^3FLAG or the MRAP2 version lacking the candidate MC4R inhibitory motif MRAP2^Δblock-3FLAG. $n = 3$ independent experiments. Data points belonging to each different experiment are encoded by translucent points of different color and shape. The average of each experiment is represented by solid points. A solid line has been used to represent the median of all combined data and dashed lines to represent the interquartile values. Asterisks indicate statistical significance values calculated by one-way ANOVA on individual cilia followed by Tukey post hoc test (n.s., nonsignificant; **** $p < 0.0001$). All underlying data are found S1 Data. (**F**) Western blot showing total MC4R^3NG and MRAP2^3FLAG protein levels in whole cell lysates of cells expressing MC4R^3NG alone or co-expressing MRAP2^3FLAG/MC4R^3NG, treated for 24 h with either HS014, AgRP, or vehicle. MC4R^3NG and MRAP2^3FLAG band densities were measured and normalized to the tubulin density (loading control). The abundances of MC4R^3NG and MRAP2^3FLAG relative to their vehicle-treated levels are indicated below the blots. AgRP, agouti-related peptide; MC4R, melanocortin receptor 4; MRAP2, melanocortin receptor accessory protein 2.

contacts made by MRAP2 with ligands may thus increase the ligand responsiveness of MC4R/MRAP2 as previously observed with α-MSH [40].

Interestingly, MRAP2 interacts with the TM5-6 helices of MC4R in the AF2 structural model. We note that TM6 was the site of extensive mutations to generate MC4R proteins amenable to biochemical purification and crystallography [48], suggesting that MRAP2 might stabilize MC4R via co-folding. However, the total protein levels of MC4R^3NG were unaffected by co-expression of MRAP2 and treatment with inverse agonists stabilized MC4R^3NG to a similar extent in the lines expressing MC4R alone or MC4R and MRAP2 (**Fig 4F**). Thus, MRAP2 does not appear to influence MC4R stability.

Given that MRAP2 does not appear to influence the tonic activity or stability of MC4R, we considered that MC4R may piggyback onto MRAP2 for entry into cilia. When stably expressed on its own, MC4R^3NG was undetectable in cilia, mirroring findings in the *Mrap2^-/-* mouse [13] (**Fig 5A**). While inverse agonists increased the ciliary levels of MC4R^3NG in MRAP2/MC4R cells, they did not promote ciliary accumulation of MC4R^3NG in the MC4R line (**Figs 5A, 5B, and S3A**), suggesting that MC4R never entered cilia in the absence of MRAP2. To further test this hypothesis, we blocked ciliary exit by depleting the essential BBSome regulator ARL6 using siRNA (**Figs 5C and S3B**). MC4R levels in cilia greatly increased when ARL6 was depleted in MC4R/MRAP2 cells but not detectably in MC4R cells (**Figs 5C, 5D, and S3C**). Together, these results indicate that ciliary entry is very inefficient when MC4R is expressed alone.

As MRAP2 efficiently targets to cilia when expressed alone ([13] and **S4A and S4B Fig**), we conclude that import of MC4R/MRAP2 into cilia relies on the ciliary targeting determinants of MRAP2. Nonetheless, the simple piggyback model of MC4R hitching a ride onto MRAP2 needs to be qualified as the levels of MRAP2 increase when it is co-expressed with MC4R (**S4A and S4B Fig**). This result suggests that the powerful cilia targeting determinants of MRAP2 either synergize with the weak determinants on MC4R or becomes better exposed when MRAP2 is in a complex with MC4R. Congruent with MRAP2 and MC4R trafficking as a complex, MRAP2 undergoes constitutive clearing from cilia in the IMCD3-[MRAP2;MC4R] cell line unless inverse agonists are added (**Fig 2A, 2D and 2E**, see also **Fig 3A and 3C**). The increased ciliary levels of MRAP2 in the presence of inverse agonists are due to the formation of a stable complex between MRAP2 and MC4R, as ciliary MRAP2 levels in absence of MC4R are not affected by AgRP or HS014 (**S4C and S4D Fig**).

Somewhat unexpectedly, transient overexpression of MRAP2 in the MRAP2/MC4R cell line led to a drastic increase in the ciliary fluorescence of MC4R^3NG (**Figs 5E, 5F, and S3D**). A similar effect was observed when MRAP2 was overexpressed in the MC4R-only cell line. These results suggest that the low levels of MRAP2 and MC4R expressed in our system may not be sufficient to drive the efficient formation of the import-competent MRAP2/MC4R

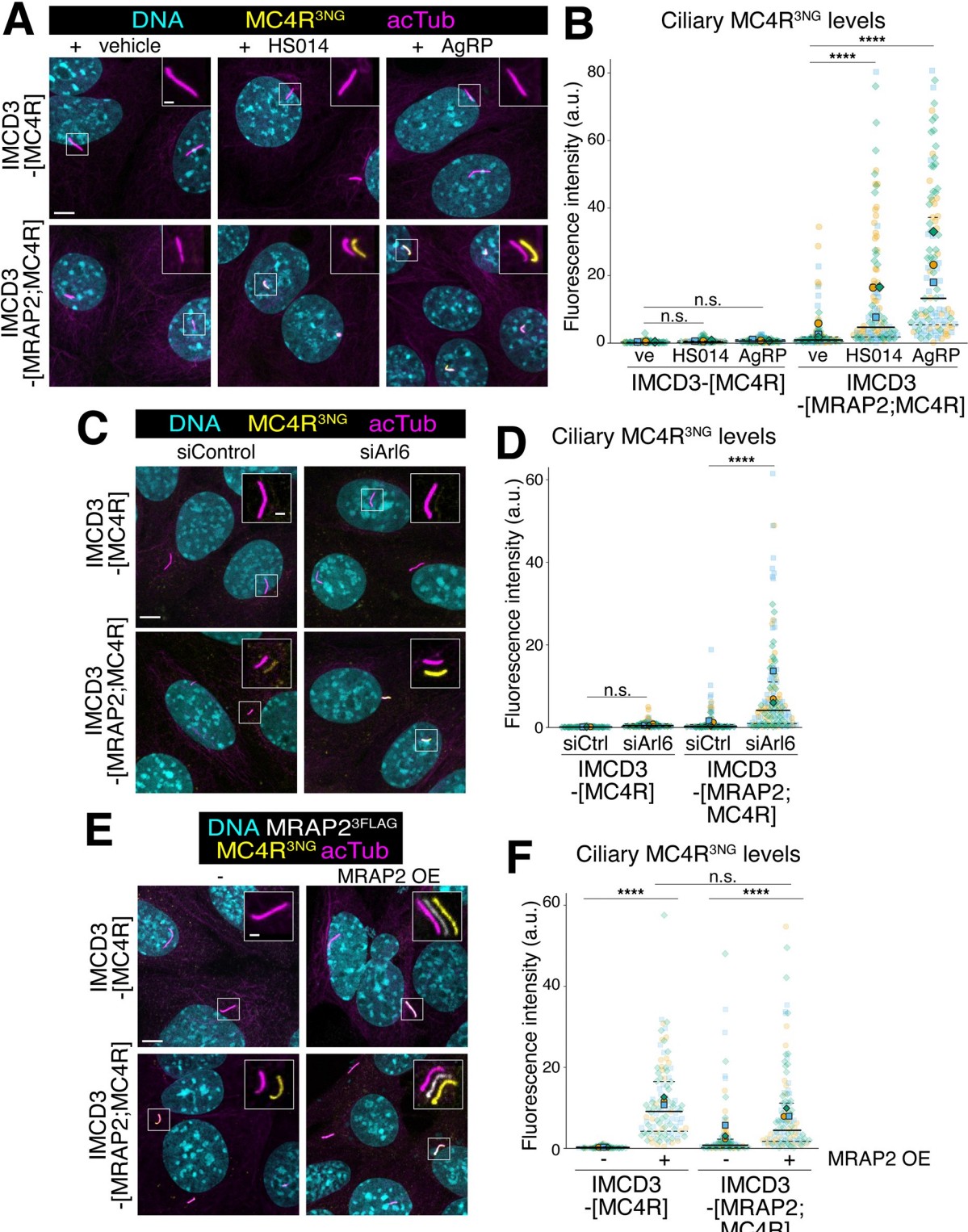

**Fig 5. MRAP2 facilitates the import of MC4R into cilia.** (**A**) Representative images of IMCD3-[MC4R^3NG] (top row) or IMCD3-[MRAP2^3FLAG;MC4R^3NG] (bottom row). Serum-starved cells were treated for 24 h with either HS014, AgRP, or vehicle, then fixed and stained for acetylated tubulin (acTub, magenta) and DNA (cyan). MC4R^3NG was visualized through the intrinsic fluorescence of NG (yellow). The yellow and magenta channels are shifted to facilitate visualization of ciliary signals in the insets. Scale bars: 5 μm (main panel) and 1 μm (inset). (**B**) Superplots of the ciliary fluorescence intensities of MC4R^3NG in IMCD3-[MC4R^3NG] and IMCD3-[MRAP2^3FLAG;MC4R^3NG]

treated with HS014, AgRP, or vehicle. $n$ = 3 independent experiments. Data points belonging to each different experiment are encoded by translucent points of different color and shape. The average of each experiment is represented by solid points. A solid line has been used to represent the median of global data and dashed lines to represent the interquartile values. Asterisks indicate statistical significance values calculated by one-way ANOVA on individual cilia followed by Tukey post hoc test (**** $p < 0.0001$). All underlying data are found S1 Data. (**C**) Representative images of IMCD3-[MC4R$^{3NG}$] (top row) or IMCD3-[MRAP2$^{3FLAG}$;MC4R$^{3NG}$] (bottom row). Cells were transfected with either siArl6 or siLuc2 (negative control, siCtrl). DNA (cyan), MC4R$^{3NG}$ (yellow), cilia (acetylated tubulin, magenta). The yellow and magenta channels have been shifted in the insets for better visualization. Scale bars: 5 μm (main panel) and 1 μm (inset). (**D**) Superplot comparing the ciliary fluorescence intensity of MC4R$^{3NG}$ in IMCD3-[MC4R$^{3NG}$] or IMCD3-[MRAP2$^{3FLAG}$;MC4R$^{3NG}$] cells, treated with either siLuc2 (negative control, siCtrl) or siArl6. $n$ = 3 independent experiments. Data points belonging to each different experiment are encoded by translucent points of different color and shape. The average of each experiment is represented by solid points. A solid line has been used to represent the median of global data and dashed lines to represent the interquartile values. Asterisks indicate statistical significance values calculated by one-way ANOVA on individual cilia followed by Tukey post hoc test (**** $p < 0.0001$). All underlying data are found S1 Data. (**E**) Representative images of IMCD3-[MC4R$^{3NG}$] (top row) or IMCD3-[MRAP2$^{3FLAG}$;MC4R$^{3NG}$] (bottom row) transiently transfected with $^{V5}$MRAP2, to distinguish the newly delivered MRAP2 from the stably expressed MRAP2$^{3FLAG}$. DNA (cyan), $^{V5}$MRAP2 (white), MC4R$^{3NG}$ (yellow), cilia (acetylated tubulin, magenta). The white, yellow, and magenta channels have been shifted in the inserts for better visualization. Scale bars: 5 μm (main panel) and 1 μm (inset). (**F**) Superplot comparing the ciliary fluorescence intensity of MC4R$^{3NG}$ in cells expressing MC4R$^{3NG}$ alone or co-expressing MRAP2$^{3FLAG}$/MC4R$^{3NG}$, transiently transfected with $^{V5}$MRAP2. $n$ = 3 independent experiments. Data points belonging to each different experiment are encoded by translucent points of different color and shape. The average of each experiment is represented by solid points. A solid line has been used to represent the median of global data and dashed lines to represent the interquartile values. Asterisks indicate statistical significance values calculated by one-way ANOVA on individual cilia followed by Tukey post hoc test (n. s., nonsignificant; **** $p < 0.0001$). All underlying data are found S1 Data. AgRP, agouti-related peptide; MC4R, melanocortin receptor 4; MRAP2, melanocortin receptor accessory protein 2.

heterodimer. Alternatively, high levels of MRAP2 may block MC4R tonic activity through an unknown mechanism.

## MC4R is continuously removed from cilia in a β-arrestin- and BBSome-dependent manner

We next investigated how tonic activity may lead to spontaneous exit of MC4R from cilia. β-arrestin 2 is a molecular sensor of the activated state of GPCRs that is required for the exit of SSTR3 and GPR161 from cilia [32,49,50]. In past work, we found that β-arrestin 2 associates with ciliary GPCRs upon their activation and directs the ubiquitination machinery to these activated ciliary GPCRs [27]. Ubiquitinated GPCRs are then recognized by the BBSome via the ubiquitin reader TOM1L2, enabling the selective removal of activated GPCRs from cilia [31]. To test whether β-arrestin 2 and the BBSome help remove the tonically active MC4R from cilia, we deleted either Arl6 ($Arl6^{-/-}$) or β-arrestins 1 and 2 ($Barr1^{-/-}/Barr2^{-/-}$) in the MRAP2/MC4R cell line. Both mutant lines displayed similar increases in the ciliary levels of MRAP2 and MC4R compared to WT cells (**Figs 6A–6C, S5A, and S5B**), suggesting that β-arrestin-dependent ubiquitination of MC4R targets activated MC4R for removal from cilia by the BBSome. Importantly, the total protein levels of MC4R increased when β-arrestin was deleted but not when $Arl6$ was deleted (**Fig 6D**). These data indicate that β-arrestin participates in the degradation of MC4R and its ciliary exit while ARL6 and the BBSome only affect ciliary exit.

## MC4R is ubiquitinated in the absence of ligand

A central prediction of the above model is that the constitutive activity of MC4R leads to a high level of ubiquitination. To test this hypothesis, we captured proteins covalently modified with ubiquitin (Ub) from cell extracts under denaturing conditions using beads coated with a high affinity ubiquitin-binding domain (UBD) [51] (**Fig 7A**). In total lysates, MC4R$^{3NG}$ was detected as a broad band, typical of glycosylated membrane proteins, centered around 130 kDa. After UBD capture, MC4R$^{3NG}$ migrated as smear of >150 kDa indicative of ubiquitin chains added to MC4R. As no MC4R ligands are present in the cell culture system, we conclude that a fraction of MC4R is constitutively ubiquitinated.

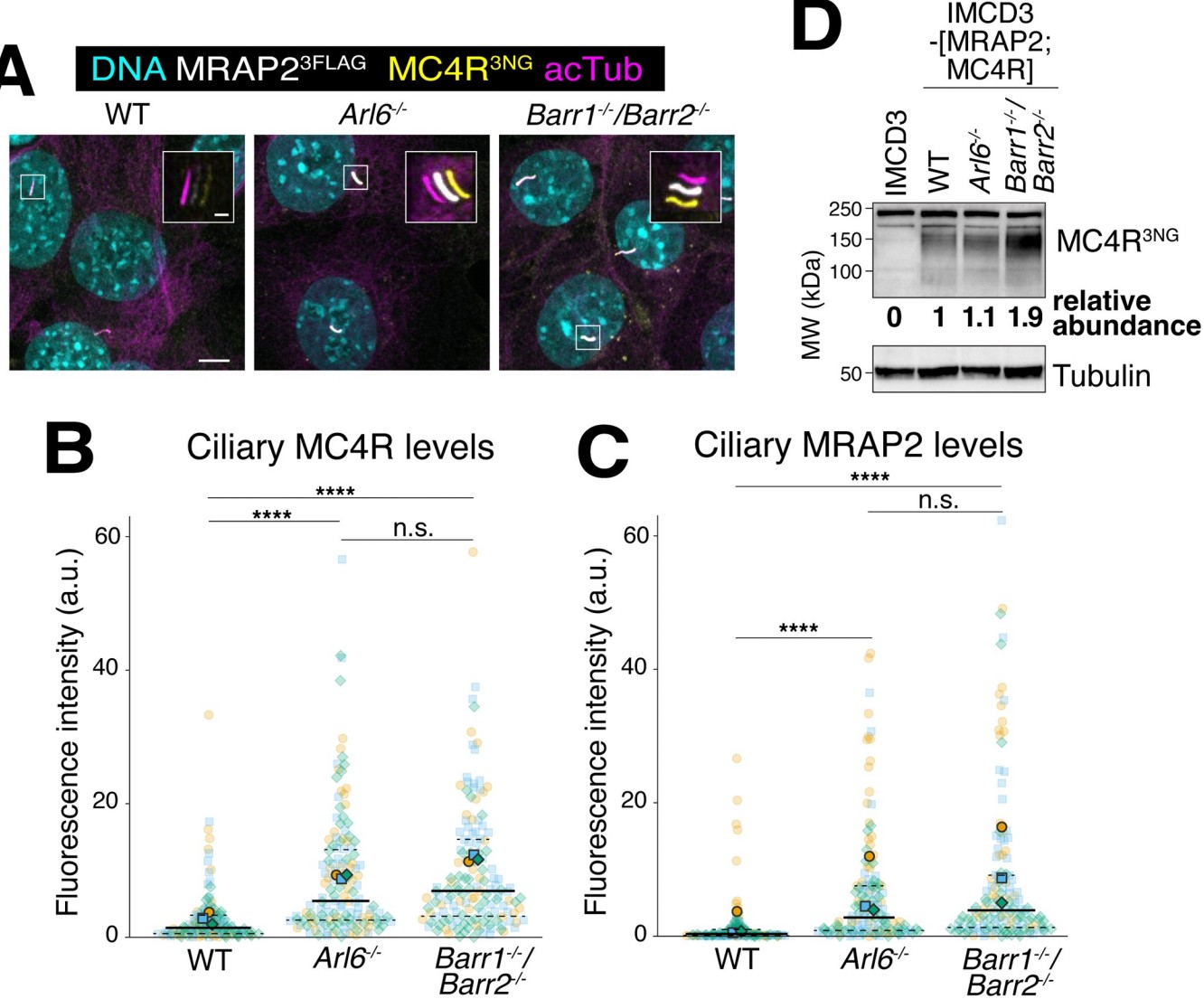

**Fig 6. Constitutive MC4R removal from cilia depends on β-arrestin and the BBSome.** (**A**) Representative images of IMCD3-[MRAP2$^{3FLAG}$;MC4R$^{3NG}$] in wild-type, *Arl6$^{-/-}$* and *Barr1$^{-/-}$/Barr2$^{-/-}$* cells. DNA (cyan), MRAP2$^{3FLAG}$ (white) MC4R$^{3NG}$ (yellow), cilia (acetylated tubulin, magenta). The white, yellow, and magenta channels have been shifted in the insets for better visualization. Scale bars: 5 μm (main panel) and 1 μm (inset). (**B**) Superplot comparing the ciliary fluorescence intensity of MC4R$^{3NG}$ in IMCD3-[MRAP2$^{3FLAG}$;MC4R$^{3NG}$] in wild-type, *Arl6$^{-/-}$* and *Barr1$^{-/-}$/Barr2$^{-/-}$* cells. *n* = 3 independent experiments. Data points belonging to each different experiment are encoded by translucent points of different color and shape. The average of each experiment is represented by solid points. A solid line has been used to represent the median of global data and dashed lines to represent the interquartile values. Asterisks indicate statistical significance values calculated by one-way ANOVA on individual cilia followed by Tukey post hoc test (n.s., nonsignificant; **** $p < 0.0001$). All underlying data are found S1 Data. (**C**) Superplot comparing the ciliary fluorescence intensity of MRAP2$^{3FLAG}$ in IMCD3-[MRAP2$^{3FLAG}$;MC4R$^{3NG}$] in wild-type, *Arl6$^{-/-}$* and *Barr1$^{-/-}$/Barr2$^{-/-}$* cells. *n* = 3 independent experiments. Data points belonging to each different experiment are encoded by translucent points of different color and shape. The average of each experiment is represented by solid points. A solid line has been used to represent the median of global data and dashed lines to represent the interquartile values. Asterisks indicate statistical significance values calculated by one-way ANOVA on individual cilia followed by Tukey post hoc test (n.s., nonsignificant; **** $p < 0.0001$). All underlying data are found S1 Data. (**D**) Western blot showing total MC4R$^{3NG}$ protein levels in whole cell lysates of IMCD3-[MRAP2$^{3FLAG}$;MC4R$^{3NG}$] in wild-type, *Arl6$^{-/-}$* and *Barr1$^{-/-}$/Barr2$^{-/-}$* cells. MC4R$^{3NG}$ band densities were measured and normalized to the tubulin density (loading control). The abundances of MC4R$^{3NG}$ and MRAP2$^{3FLAG}$ relative to the wild-type levels are indicated below the blots. MC4R, melanocortin receptor 4; MRAP2, melanocortin receptor accessory protein 2.

Ub chains are assembled by conjugating successive Ub molecules onto one of 7 lysines or on the amino terminus of Ub. Each linkage dictates a specific biological outcome, with K11 and K48 linkages targeting soluble proteins to the proteasome and K63 linkages functioning

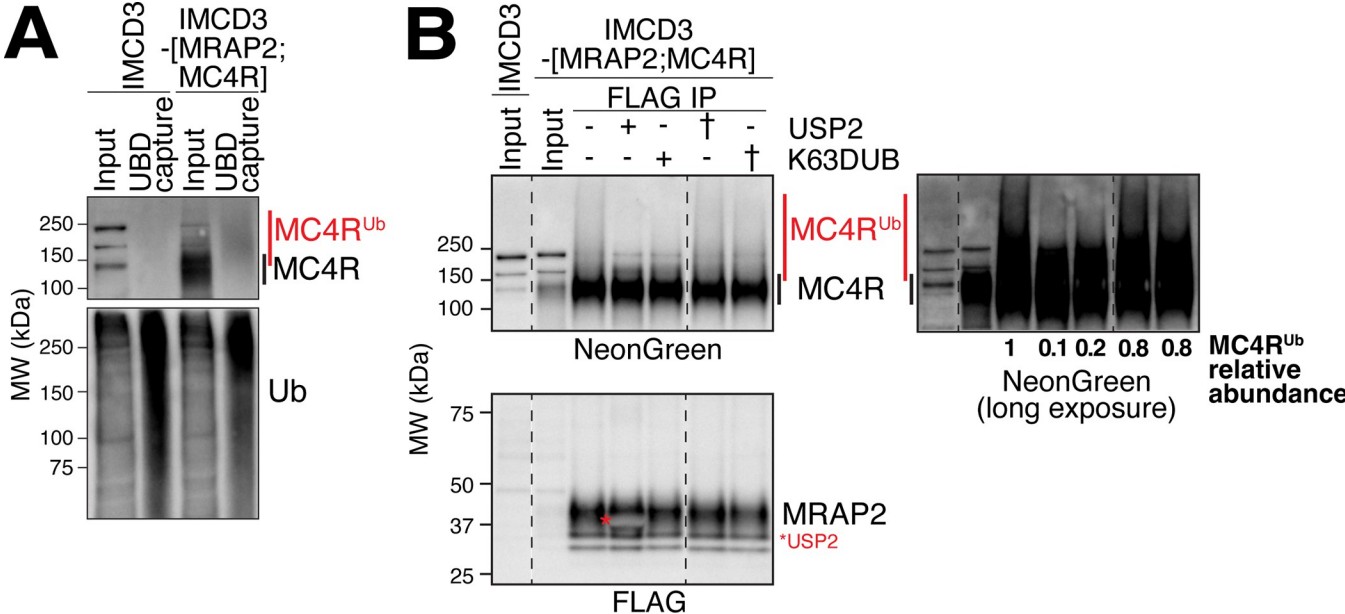

**Fig 7. MC4R is K63-polyubiquitinated in the absence of ligand.** (**A**) Western blot of UBD-captured proteins in IMCD3-[MRAP2[3FLAG];MC4R[3NG]] and FlpIn parental cells. The top panel shows MC4R[3NG] detected with anti-NeonGreen antibody. A band below 150 kDa (black bracket) that is absent in the parental lanes can be seen in the IMCD3-[MRAP2[3FLAG];MC4R[3NG]] input lane. MC4R is detected as a higher molecular smear above 150 kDa in the UBD-captured lane (red bracket). The bottom panel shows the efficiency of the ubiquitinated protein capture as increased smears of ubiquitin. (**B**) Western blot of MRAP2[3FLAG]-captured proteins in IMCD3-[MRAP2[3FLAG];MC4R[3NG]] and FlpIn parental cells. MC4R[3NG] co-immunoprecipitates with MRAP2 and appears as a band below 150 kDa (black bracket) and a higher molecular smear that corresponds to polyubiquitinated MC4R (red brackets). The higher molecular smear disappears when the eluates are treated with the K63-specific deubiquitinase AMSH (K63DUB) and the pan-linkage deubiquitinase USP2. The smear stays intact when the deubiquitinases are heat-inactivated (†) prior to the eluate treatment. An overexposed anti-NeonGreen blot is shown to better visualize the smears. The asterisk points to the band of USP2 added protein that overlaps with the MRAP2[3FLAG] band. MC4R, melanocortin receptor 4; MRAP2, melanocortin receptor accessory protein 2; UBD, ubiquitin-binding domain.

chiefly in endolysosomal sorting [52]. We and others have shown that K63-linked ubiquitin (K63Ub) chains added onto ciliary GPCRs target them for ciliary exit [27,28]. To test whether the Ub chains added onto MC4R are K63-linked, we leveraged ubiquitin chain restriction using deubiquitinases with defined specificities [53]. Upon immunoprecipitation of the MRAP2/MC4R complex, we detected a high molecular weight smear above the main MC4R band suggestive of polyubiquitinated MC4R (**Fig 7B, third lane**). The disappearance of this high molecular weight smear upon treatment with the pan-linkage deubiquitinase USP2 or the K63-specific deubiquitinase STAMBP/AMSH (thereafter referred to as K63DUB, **Fig 7B, fourth and fifth lanes**) established that the chains built onto MC4R in the absence of ligand are K63-linked. Heat inactivation of the enzymes preserved the smear (**Fig 7B, sixth and seventh lanes**). To functionally test the role of K63Ub chains in ciliary exit of MC4R, we targeted the catalytic domain of K63DUB to the cilium. Transient expression of cilia-K63DUB resulted in the marked accumulation of MC4R in cilia (**Figs 8A, 8B, and S6A**) but expression of a cilia-targeted catalytically dead K63DUB (K63DUB[†]) did not elevate the ciliary levels of MC4R. We conclude that MC4R is constitutively marked with K63Ub chains and that K63Ub-modified MC4R exits cilia.

To test whether the K63 ubiquitin chains added onto MC4R trigger its exit from cilia, we mutated all 7 cytoplasm-facing lysine residues (cK) in MC4R to arginine (**Fig 8C**). The resulting ubiquitination-resistant variant MC4R$_{cK0}$ accumulated in cilia at a higher level than MC4R (**Figs 8C, 8D, and S6B–S6D**). Yet, ciliary levels of MC4R$_{cK0}$ were somewhat lower than MC4R levels in cells expressing cilia-K63DUB, suggesting that the existence of further Ub acceptor

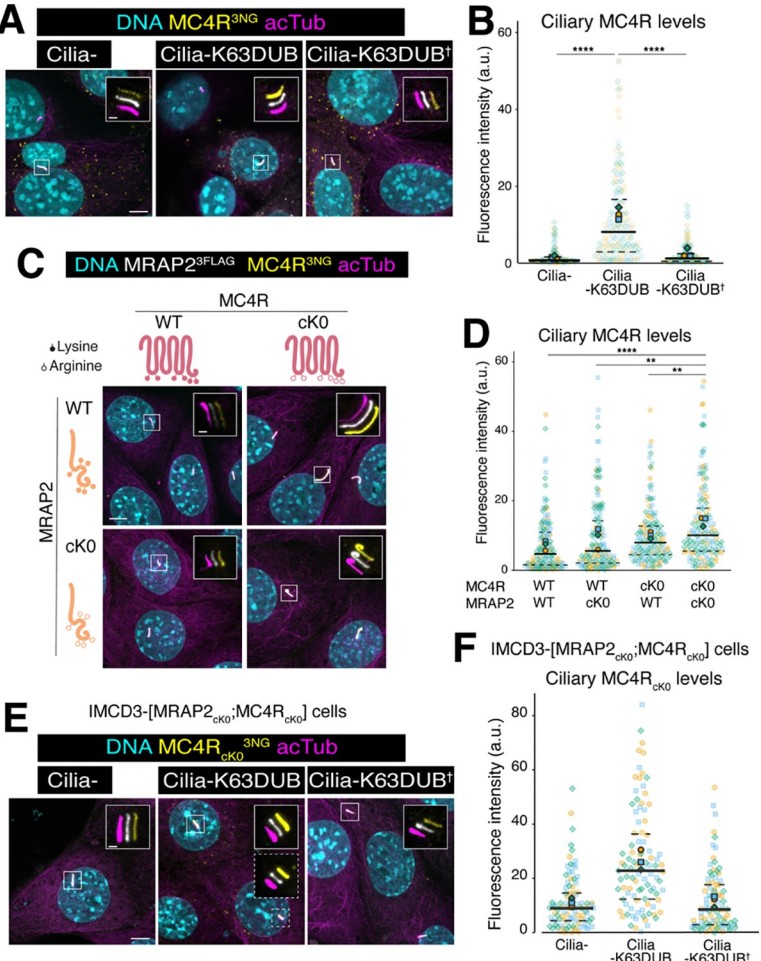

**Fig 8. Acceptor lysines required for ciliary exit are distributed on MC4R and its ciliary partners.** (**A**) Representative images of IMCD3-[MRAP2$^{3FLAG}$;MC4R$^{3NG}$] cells transiently transfected with cilia-targeted K63-specific deubiquitinase catalytic domain of AMSH (cilia-K63DUB), the catalytically inactive counterpart (cilia-K63DUB$^†$) or the cilia targeting sequence alone fused to the mScarlet reporter (cilia-) used for all constructs. DNA (cyan), transiently transfected construct (white), MC4R$^{3NG}$ (yellow), cilia (acetylated tubulin, magenta). The white, yellow, and magenta channels have been shifted in the insets for better visualization. Scale bars: 5 μm (main panel) and 1 μm (inset). (**B**) Superplot comparing the ciliary fluorescence intensity of IMCD3-[MRAP2$^{3FLAG}$;MC4R$^{3NG}$] cells transiently transfected with cilia-targeted K63-specific deubiquitinase catalytic domain of AMSH (cilia-K63DUB), the catalytically inactive counterpart (cilia-K63DUB$^†$) or the cilia targeting sequence alone fused to the mScarlet reporter (cilia-). $n = 3$ independent experiments. Data points belonging to each different experiment are encoded by translucent points of different color and shape. The average of each experiment is represented by solid points. A solid line has been used to represent the median of global data and dashed lines to represent the interquartile values. Asterisks indicate statistical significance values calculated by one-way ANOVA on individual cilia followed by Tukey post hoc test (**** $p < 0.0001$). All underlying data are found S1 Data. (**C**) Serpentine schematic representing the location of the lysine-to-arginine substitutions (lysine appears as a filled circle, whereas arginine appears as an empty circle) in MC4R$^{3NG}$ (pink) and MRAP2$^{3FLAG}$ (orange). Representative images of non-clonal IMCD3 lines stably co-expressing combinations of wild-type MRAP2$^{3FLAG}$/MC4R$^{3NG}$ (WT) or the ubiquitination-refractory variants where the intracellular lysines have been substituted by arginine residues (cK0). DNA (cyan), MRAP2$^{3FLAG}$ (white), MC4R$^{3NG}$ (yellow), cilia (acetylated tubulin, magenta). The white, yellow, and magenta channels have been shifted in the insets for better visualization. Scale bars: 5 μm (main panel) and 1 μm (inset). (**D**) Superplot comparing the ciliary fluorescence intensity of MC4R$^{3NG}$ in non-clonal IMCD3 lines stably co-expressing wild-type MRAP2$^{3FLAG}$/MC4R$^{3NG}$ (WT, serpentine schematic without empty circles), or the ubiquitination-refractory variants where the intracellular lysines have been substituted by arginine residues (K0, serpentine schematic with empty circles indicating the location of the lysine-to-arginine substitution). $n = 3$ independent experiments. Data points belonging to each different experiment are encoded by translucent points of different color and shape. The average of each experiment is represented by solid points. A solid line has been used to represent the median of global data and dashed lines to represent the interquartile values. Asterisks indicate statistical significance values calculated by one-way ANOVA on individual cilia followed by Tukey post hoc test (** $p < 0.005$; **** $p < 0.0001$). (**E**) Representative images of IMCD3-

[MRAP2$_{cK0}$$^{3FLAG}$;MC4R $_{cK0}$$^{3NG}$] cells transiently transfected with the cilia-targeted K63-specific deubiquitinase catalytic domain of AMSH (cilia-K63DUB), the catalytically inactive counterpart (cilia-K63DUB$^†$) or the cilia targeting sequence alone fused to the mScarlet reporter (cilia-). DNA (cyan), transiently transfected construct (white), MC4R$^{3NG}$ (yellow), cilia (acetylated tubulin, magenta). The white, yellow, and magenta channels have been shifted in the insets for better visualization. Scale bars: 5 μm (main panel) and 1 μm (inset). All underlying data are found S1 Data. (**F**) Superplot comparing the ciliary fluorescence intensity of IMCD3-[MRAP2$_{cK0}$$^{3FLAG}$;MC4R $_{cK0}$$^{3NG}$] cells transiently transfected with cilia-targeted K63-specific deubiquitinase catalytic domain of AMSH (cilia-K63DUB), the catalytically inactive counterpart (cilia-K63DUB$^†$) or the cilia targeting sequence alone fused to the mScarlet reporter (cilia-). $n = 3$ independent experiments. Data points belonging to one individual experiment are encoded by translucent points of a specific color and shape. The average of each experiment is represented by solid points. A solid line has been used to represent the median of global data and dashed lines to represent the interquartile values. Asterisks indicate statistical significance values calculated by one-way ANOVA on individual cilia followed by Tukey post hoc test (**** $p < 0.0001$). All underlying data are found S1 Data. MC4R, melanocortin receptor 4; MRAP2, melanocortin receptor accessory protein 2.

sites. Mutating of all 6 cytoplasmic lysines to arginines in MRAP2 (MRAP2$_{cK0}$, **Fig 8C**) significantly increased the ciliary levels of MC4R and MRAP2 beyond those in MC4R$_{cK0}$/MRAP2 cells (**Figs 8C, 8D, and S6B–S6D**). However, co-expression of MRAP2$_{cK0}$ with MC4R only modestly altered the ciliary levels of MRAP2 and MC4R, indicating that MRAP2 is a minor Ub acceptor when ubiquitination sites are available on MC4R.

To determine whether there are additional ubiquitination targets besides MC4R and MRAP2, we introduced cilia-K63DUB into MC4R$_{cK0}$/ MRAP2$_{cK0}$ cells. Remarkably, cilia-K63DUB further increased the ciliary levels of MC4R in MC4R$_{cK0}$/ MRAP2$_{cK0}$ cells (**Fig 8E and 8F**), indicating the existence of additional Ub acceptor sites beyond MC4R and MRAP2. We conclude that the cytoplasmic lysines of MC4R are primarily responsible for its constitutive exit from cilia and that the cytoplasmic lysines of MRAP2 and of other interaction partners of MC4R offer additional ubiquitination sites supporting exit of the MC4R/MRAP2 complex from cilia.

## Discussion

### MRAP2 acts mainly by promoting MC4R ciliary import

MRAPs interact with select melanocortin receptors to modulate several aspects of their activity. We previously demonstrated that MRAP2 interaction with MC4R is essential for the localization of MC4R to cilia [13], but the underlying mechanism remained to be determined. We now find that MRAP2 is not required to stabilize MC4R whether ligands are present or not and that MRAP2 does not appear to block the tonic exit of MC4R from cilia. Instead, MC4R relies on the ciliary targeting determinants of MRAP2 for entry into cilia. The ciliary targeting signal encoded within MRAP2 and the machinery that imports MRAP2/MC4R into cilia remain to be elucidated.

### Constitutive MC4R activity drives its ciliary exit

In this study, we demonstrate that the tonic activity of MC4R drives its constitutive ubiquitination and exit from primary cilia (**Fig 9**). MC4R exit from cilia is dependent on β-arrestin and on the BBSome. β-arrestin is a sensor of the active state of GPCRs and is likely required to recruit a ubiquitin ligase to tonically active MC4R, given that β-arrestin is also required for activation-dependent ubiquitination of SSTR3 and GPR161 in cilia [27]. While the identity of the ubiquitin ligase that builds ubiquitin chains on MC4R in cilia remains to be determined, the K63-linked ubiquitin chains attached to MC4R are functionally important for MC4R exit from cilia as cleavage of K63-linked ubiquitin chains inside cilium increases the ciliary levels of MC4R.

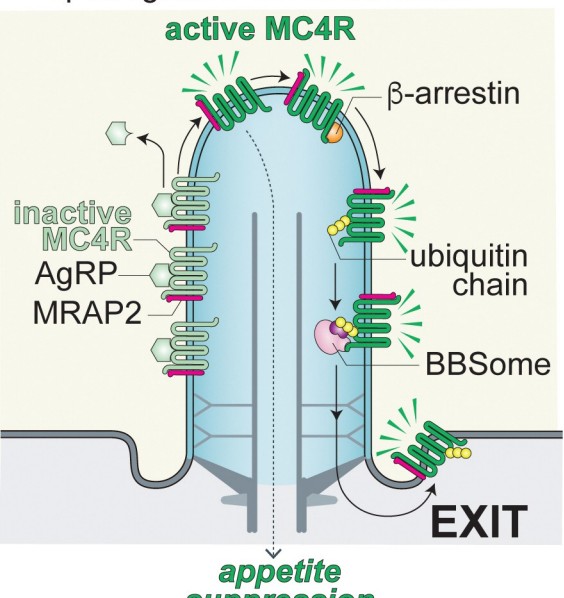

**Fig 9. General and MC4R-specific models of regulated trafficking.** Most ciliary GPCRs undergo exit once activated by their agonist. In the case of MC4R, exit is spontaneous in the absence of ligand and only in the presence of an inverse agonist does MC4R exit become interrupted. In both models, β-arrestin senses the activated state of the GPCR (agonist bound for most GPCRs, ligand free or agonist-bound for MC4R) and recruits the ubiquitination machinery to the activated GPCR. Unique to MC4R is that the receptor constitutively couples to β-arrestin. Once ubiquitinated, ciliary GPCRs are committed for retrieval back into the cell via TOM1L2 (dark purple) which endows the BBSome (pink) with the ability to recognize ubiquitin chains and move ubiquitinated proteins out of cilia. AgRP, agouti-related peptide; GPCR, G protein-coupled receptor; MC4R, melanocortin receptor 4; MRAP2, melanocortin receptor accessory protein 2.

Somewhat unexpectedly, ubiquitin acceptors sites are not only distributed on MC4R but instead extend onto MRAP2 and other associated proteins. As the ubiquitinated entity recognized by the BBSome/TOM1L2 is likely to be a complex of MC4R, MRAP2, and β-arrestin (see below), it is conceivable that β-arrestin offers additional ubiquitin acceptor sites for the ciliary ubiquitin ligase that recognizes the active conformation of MC4R. A striking precedent is provided by ubiquitination of β-arrestin driving endocytosis of associated activated GPCRs [54,55].

The constitutive ciliary exit of MC4R is blocked when MC4R is treated with its endogenous ligand AgRP or with the synthetic ligand HS014. While AgRP and HS014 can function as antagonists to α-MSH [38,56], MC4R agonists are absent from our cell culture system. Instead, AgRP and HS014 are known to decrease the tonic activity of MC4R in bulk cAMP assays, making them inverse agonists in addition to antagonists [36,37]. This suppression of MC4R exit by inverse agonists leads us to propose that the high tonic activity of MC4R is responsible for its constitutive ubiquitination and exit. The increase in ciliary MC4R levels upon inhibition of tonic activity is recapitulated in mice with genetic or physiological manipulations that result in elevated AgRP levels (Vaisse lab, in preparation). The findings in our cell culture system where MC4R ligands are precisely set and where no synaptic input is present, affirm that suppression of MC4R exit by inverse agonism is MC4R- and cell-autonomous. Further highlighting the physiological relevance of our findings, the pronounced effect of AgRP on ciliary accumulation of MC4R even in the presence of vast excesses of α-MSH provides cell-based evidence for the highly potent antagonist activity of AgRP, which is best exemplified in vivo by the lasting hyperphagia that follows intracerebroventricular injection of nanomolar concentrations of AgRP [57].

### The molecular logic of regulated GPCR exit from cilia

While tonically active MC4R spontaneously exits cilia, other tonically active GPCRs do not undergo spontaneous exit from cilia. The ciliary GPCR serotonin receptor 6 (5-HT$_6$R) possesses a high level of tonic activity [58,59], yet 5-HT$_6$R is robustly detected in cilia [60,61]. Similarly, GPR161 is a self-activating GPCR that only exits cilia when SMO becomes activated [62,63]. Given that 5-HT$_6$R, GPR161, and MC4R all tonically activate Gα$_S$ in the absence of ligands, yet only MC4R undergoes constitutive exit from cilia, Gα$_S$ activation is not the primary determinant of GPCR exit from cilia.

Instead, the assembly of a stable complex between β-arrestin and an activated GPCR is the most likely intermediate that triggers regulated exit from cilia (**Fig 9**). The formation of a stable GPCR/β-arrestin complex requires phosphorylation of the C-tail of the GPCR by GPCR kinases (GRKs) which are activated by active GPCRs [64,65]. Strong coupling to GRKs is thus likely to represent a major determinant for ciliary exit of a given GPCR. MC4R is unusual among GPCRs in that its constitutive activity results in both Gα$_S$ activation and β-arrestin recruitment, whereas tonically active GPCRs typically only activate Gα$_S$ and do not stably associate with β-arrestin [66]. Importantly, it has been shown that AgRP suppresses tonic β-arrestin 2 binding to MC4R in a dose-dependent manner [67], thus supporting our model that MC4R constitutively engages β-arrestin to promote its exit from cilia unless AgRP is present.

We note that the MC4R[V103I] variant, which protects against obesity, displays increased binding to β-arrestin [67,68]. How increased binding to β-arrestin may ameliorate the physiological activity of MC4R is paradoxical given that such an increase is predicted to further depress the ciliary levels of MC4R, and MC4R signals within cilia [11]. It is thus conceivable that, in addition to recruiting an ubiquitin ligase to activated MC4R, β-arrestin is activated by MC4R within cilia where it may signal to downstream signaling partners. Separating the roles of β-arrestin in signaling and in MC4R exit will require the identification of the ubiquitin ligase that catalyzes the activity- and β-arrestin-dependent ubiquitination of MC4R.

### Physiological importance of activity-dependent MC4R exit from cilia

We have previously demonstrated that MC4R localization to primary cilia, in particular in PVN neurons, is required for its function. Specifically, we have demonstrated that genetically ablating cilia or inhibiting cAMP production in the cilia of these MC4R neurons results in increased food intake and obesity [13]. Moreover, several MC4R mutations found in human patients specifically impact its targeting to primary cilia [11].

Thus, in the context of cilium-based MC4R signaling, the continuous exit of MC4R from cilia appeared paradoxical. Yet, one needs to consider that evolutionary pressures have selected mechanisms that increase food intake to maximize survival of the organism. Thus, the apparent paradox of MC4R signaling from within cilia while keeping its ciliary levels very low can be viewed as a necessary adaptation to the survival of the organism as, without constitutive coupling to β-arrestin, tonically active MC4R would remain inside cilia and potently suppress food intake, leading to starvation and eventual death of the organism.

## Material and methods

### Cell culture

IMCD3 cells were cultured in DMEM/F12 (11330–057; Gibco) supplemented with 10% FBS (Gemini Bio-products), 100 U/ml penicillin-streptomycin (Gemini Bio-products), and 2 mM L-glutamine (Gemini Bio-products), on acid-washed 12 mm #1.5 cover glass (Fisherbrand, Thermo Fisher Scientific). Transfection and/or treatments are specified for each experiment.

Ciliation was induced for 24 h prior to fixation by serum starvation in media containing 0.2% FBS for 24 h, or media completely devoid of serum for ligand treatment.

## Plasmid construction and stable line generation

Stable isogenic IMCD3 cell lines were generated using the Flp-In system (Thermo Fisher Scientific) as previously described [69]. To generate the main mMRAP2^3FLAG-T2A-hMC4R^3NG construct, we amplified the murine MRAP2 sequence NM_001359955.1 from the third codon, a second methionine residue, and the human MC4R sequence NM_005912.3 from previously published plasmids [11,13]. The T2A sequence was added via PCR primers. The mMRAP2^3FLAG-T2A-hMC4R^3NG expression cassette was cloned into a modified pEF5/FRT (flippase recognition target) plasmid where expression is driven by an attenuated pEF1α promoter lacking the TATA box (pEF1α^A) and the antibiotic resistance switched to Blasticidine. MRAP2/MC4R was introduced into WT IMCD3-FlpIn cells and in *Barr1*^-/-/*Barr2*^-/- IMCD3-FlpIn cells [70].

The *Arl6*^-/- MRAP2/MC4R line was generated by CRISPR-mediated knock-out of *Arl6* in the wild-type MRAP2/MC4R line. Cas9 and guide RNA (GCCTGGGGCTGGATAACAGT) were transiently expressed from a pX459 derivative and transfectants selected with puromycin. Clones were isolated by limited dilution and selected by western blotting. The genotype (NM_001347244.1:c.79_89del; c.56_111del) was determined by amplification of the targeted DNA region, DNA sequencing, and DECODR analysis (Deconvolution of Complex DNA Repair).

The GPR161^3NG line used as a calibrator for measuring the absolute number of MC4R^3NG molecules per cilia was described in [70] and the molecular calibration in [32]. The absolute number of ciliary MC4R^3NG molecules was estimated by linearly interpolating the mean fluorescence value of all measured single cilia in the MRAP2/MC4R line ($n = 186$), based on our previous calculation that the ciliary mean fluorescence value of the GPR161^3NG line ($n = 184$ cilia in this study) is equivalent to 1,226 molecules.

We inferenced the location of intracellularly exposed lysine residues from the recently published structural data on MC4R [48] and from the AlphaFold3 model of MC4R/MRAP2, and mutated them to arginines. The mutations in MC4R_cK0 are K71R/K73R/K224R/K242R/K311R/K314R and the mutations in MRAP2_cK0 are K71R/K98R/K106R/K132/K165/K196. Plasmids harboring mMRAP2_cK0^3FLAG-T2A-hMC4R^3NG, mMRAP2^3FLAG-T2A-hMC4R_cK0^3NG, and mMRAP2_cK0^3FLAG-T2A-hMC4R_cK0^3NG were gene synthesized (GenScript) and stable cell pools were generated (alongside a new non-clonal mMRAP2^3FLAG-T2A-hMC4R^3NG for comparison) by co-electroporation with a plasmid encoding the Flp recombinase (pOG44), using a Neon electroporator (Thermo). Transformants were selected by blasticidin resistance (4 μg/ml) followed by FACS sorting.

We also identified a short helical segment from the MRAP2 cytoplasmic tail that inserts into the Gα binding site, spanning from amino acids E84 to G97. We removed this segment to generate MRAP2^Δblock-3FLAG. Briefly, the original MRAP2^3FLAG vector [13] was restricted with EcoNI, and the portion encoding from E84 to G97 was seamlessly removed using Gibson-mediated repair. Frame was checked by Sanger sequencing.

Cilia-K63DUB was generated by fusing the catalytic domain of mouse AMSH (gift from David Komander, Walter and Eliza Hall Institute of Medical Research, Melbourne, Australia: plasmid no. 66712; Addgene; [71]) with NPHP3[1–200] and mScarlet to create NPHP3[1–200]-mScarlet-AMSH[243–424]. The catalytically dead version of NPHP3[1–200]-mScarlet-AMSH^†[243–424] was generated by mutating the catalytic residue Glu280 to Ala [72]. The AMSH catalytic domain was removed from NPHP3[1–200]-mScarlet-AMSH[243–424] and substituted by a stop codon to generate the ciliary targeted control.

## Transient transfection

Cells were transfected using X-tremeGENE 9 DNA Transfection Reagent (Roche). The transfection reagent was diluted in OptiMEM (Invitrogen) and incubated at room temperature for 5 min. Then, the mixture was added to the diluted plasmids in a 6:1 ratio (6 µl transfection reagent to 1 µg DNA) and incubated at room temperature for 20 min. In total, 50,000 cells in suspension were added to the transfection mixture in a 24-MW plate. Transfected cells were switched to starvation medium after 24 h and fixed 16 h afterwards as described below for immunofluorescence.

## Quantitative RT-PCR

IMCD3-[mMRAP2$^{3FLAG}$-T2A-hMC4R$^{3NG}$], IMCD3-[GPR161$^{3NG}$], and parental IMCD3-cells were plated onto 10 cm plates at a density of 1 million cells/plate and allowed to grow for 48 h in DMEM/F12 supplemented with 10% FBS. Total RNA was extracted using the Quick-RNA Miniprep kit (Zymo Research) following the manufacturer's instructions. Purity of total RNA was determined as 260 nm/280 nm absorbance ratio with expected values between 1.8 and 2.00 by the NanoDrop spectrophotometer (Thermo Scientific), and 1 µg of extracted total RNA was reversed transcribed using iScript cDNA Synthesis kit (Bio-Rad).

Quantitative RT-PCR was performed in a 96-well format in a C1000 TouchThermal cycler with CFX96 Touch Real Time PCR Detection System (Bio-Rad). The PCR mixtures consisted of 2.5 µl cDNA corresponding to approximately 12.5 ng total RNA, 0.25 µm of primers, 5 µl Sso Fast EvaGreen Supermix (Bio-Rad) in a final volume of 10 µl. The assay included no template and RT minus controls to detect reagent contamination and presence of genomic DNA. The thermal profile of the RT-PCR procedure had an initial denaturation step of 95˚C for 30 s, followed by 40 cycles of: (1) 95˚C for 5 s for denaturation; (2) 60˚C for 25 s for annealing and extension (amplification data collected at the end of each cycle). Dissociation curves were used to validate product specificity. All samples were amplified in triplicates from the same total RNA preparation and the mean value of these triplicates was used for further analysis. Relative fold-change was calculated using the $2^{-\Delta\Delta CT}$ method.

A primer pair targeting exclusive binding sites within the 3xmNeonGreen fusion and the linker common to MC4R$^{3NG}$ and GPR161$^{3NG}$ was designed. Validated primer pairs for housekeeping genes *Gapdh* and *Bact* were used [73]. The relative expression of each gene was normalized to *Gapdh*, as it was the best control. Primer sequences are: GTGGTTGATATCCAG CACAGTGG and GTTATCCTCCTCTCCTTTGGAAACC for 3xmNeonGreen and AGGTC GGTGTGAACGGATTTG and TGTAGACCATGTAGTTGAGGTCA for *Gapdh*.

## Immunofluorescence

Cells were fixed in phosphate-buffered saline (PBS) containing 4% paraformaldehyde (Electron Microscopy Sciences) for 15 min at 37˚C. Cells were then permeabilized in PBS containing 0.1% Triton X-100 (BP151-500, Thermo Fisher Scientific), 5% normal donkey serum (017-000-121, Jackson ImmunoResearch), and 3% BSA (BP1605-100, Thermo Fisher Scientific) for 30 min. Permeabilized cells were incubated with primary antibodies (see **S1 Table**) for 1 h, washed with PBS, and incubated with dye-coupled secondary antibodies for 30 min. Cells were then washed with PBS, stained with Hoechst 33342 DNA dye (Invitrogen), and washed with PBS before mounting with Fluoromount G (Electron Microscopy Sciences). 3xmNeon-Green-tagged proteins were detected via their intrinsic fluorescence.

All micrographs displayed in the figure panels, except for Figs 3B and S4, were acquired with an Airyscan LSM 900 (Zeiss) microscope, using Z stacks of 12–18 planes with 0.2 µm separation between planes. For quantification of fluorescence signals and display in Figs 3B and S4, images

were captured on a widefield fluorescence DeltaVision microscope (Applied Precision) equipped with a PlanApo 60×/1.40NA objective lens (Olympus), a pco.edge 4.2 sCMOS camera, a solid state illumination module (Insight), and a Quad polycroic (Chroma). Z stacks with 0.2 μm separation between planes were acquired using SoftWoRx. The illumination settings were adjusted to avoid signal saturation in each experimental setup, except for the imaging of hMC4R$^{3NG}$, which was kept constant throughout all experiments: 222 μW 475 nm wavelength for 0.5 s per slice. Images were flat field corrected and maximally projected using Fiji. A mask of 3 pixels width was drawn along each measured cilium. Background-corrected ciliary fluorescence intensities ($F_{cilia}$) were calculated by subtracting from the raw integrated density of cilia fluorescence ($F_{raw}$) the integrated density of an adjacent cell area ($F_{background}$). Ciliary intensities ($F_{cilia} = F_{raw} - F_{background}$) of at least 50 cilia per condition (30 for experiments involving transient transfection) were collected in 3 independent experiments and plotted using SuperPlots of Data [74]. Statistical analyses were performed on the combined data from all 3 experiments using GraphPad Prism.

### Immunofluorescence in nonpermeabilized cells

Cells were fixed in PBS containing 4% paraformaldehyde (Electron Microscopy Sciences) for 15 min at 37˚C. Cells were then blocked in the absence of detergent with PBS containing 5% normal donkey serum (017-000-121, Jackson ImmunoResearch) and 3% BSA (BP1605-100, Thermo Fisher Scientific) for 30 min. Cells were incubated with anti-FLAG M2 diluted in PBS/3%BSA for 1 h, washed with PBS, and incubated with dye-coupled secondary antibodies diluted in PBS/3%BSA for 30 min. Cells were then washed with PBS, stained with Hoechst 33342 DNA dye (Invitrogen), and washed with PBS before mounting with Fluoromount G (Electron Microscopy Sciences). 3xmNeonGreen-tagged proteins were detected via their intrinsic fluorescence. Images were captured on a widefield fluorescence DeltaVision microscope (Applied Precision) as described above.

### Ligand treatments

ddH$_2$O-diluted Human AgRP (003–53; Phoenix Pharmaceuticals) or an equivalent volume of ddH$_2$O were added 16 h after starvation to a final concentration of 50 nM. ddH$_2$O-diluted HS014 (1831; Tocris) or an equivalent volume of ddH$_2$O were added 16 h after starvation to a final concentration of 5 nM. α-MSH (2584; Tocris) was diluted in ddH$_2$O and added to the cells after starvation to final concentrations between 10 and 1,000 nM (**Fig 3**).

### Antibodies

mNeonGreen antibodies were generated by injecting rabbits with recombinant His-mNeon-Green and affinity purified on GST-mNeonGreen columns using standard procedures. All other antibodies are listed in **S1 Table**.

### Biochemical analysis of MC4R ubiquitylation

IMCD3-[mMRAP2$^{3FLAG}$-T2A-hMC4R$^{3NG}$] cells were plated onto 15 cm plates at a density of 3 million cells/plate and allowed to grow for 24 h in DMEM/F12 supplemented with 10% FBS. Cells were starved in serum-free medium to promote ciliation, and 18 h after medium change, cells were washed with ice-cold 1× PBS and scraped off on ice into 1.5 ml ice-cold immunoprecipitation buffer (50 mM HEPES (pH 7.5), 250 mM NaCl, 2 mM EDTA, 10% glycerol, 1 mM Na$_3$O$_4$V, 1 mM NaF, 1 mM N-ethylmaleimide, 0.25% w/v n-dodecyl β-D-maltoside) supplemented with protease inhibitors (1 mM 4-(2-aminoethyl)benzenesulfonyl fluoride

hydrochloride, 0.8 mM aprotinin, 15 mM E-64, 10 mg/ml bestatin, 10 mg/ml pepstatin A, and 10 mg/ml leupeptin). After incubation on ice for 10 min, lysates were clarified by centrifugation in a tabletop Eppendorf centrifuge at $20,000 \times g$ and 4°C for 15 min, followed by centrifugation in a Beckman Optima MAX-XP ultracentrifuge equipped with a TLA55 rotor at 42,000 rpm ($100,000 \times g$) at 4°C for 1 h. Clarified lysates were incubated with 20 μl anti-FLAG M2 affinity gel slurry (A2220, Sigma-Aldrich) for 2 h at 4°C to capture mMRAP2[3FLAG]. Beads were washed 5 times with 500 μl immunoprecipitation buffer and twice with mock elution buffer (1× TBS with 25 mM KCl and 5 mM $MgCl_2$). Bound proteins were eluted by antibody competition with 150 μg/ml 3xFLAG peptide (F4799 Sigma-Aldrich) in ice-cold elution buffer twice on ice for 30 min. Eluates were treated with purified 2 μm USP2cc and 250 nM AMSH [71,75] at room temperature and 37°C, respectively, for 1 h. Proteins were concentrated by methanol/chloroform precipitation and resuspended in 1× lauryl dodecyl sulfate sample loading buffer. Proteins were resolved by SDS-PAGE and transferred onto polyvinylidene fluoride membranes, and membranes were probed with anti-NeonGreen and anti-FLAG M2 antibodies. Signals were acquired with a ChemiDoc imager (Bio-Rad) and analyzed with ImageLab 6.0.1 (Bio-Rad). The integrated densities of the smears above MC4R signal were normalized to the integrated densities of the main MC4R band to correct for variations in the recovery of proteins on the affinity gel. The values were background-corrected by subtracting the value of the equivalent area in the IMCD3 control lane.

## Capture of ubiquitinated proteins

The capture of ubiquitinated proteins from crude cell lysates was performed under denaturation conditions as described in [51]. Briefly, IMCD3-[mMRAP2[3FLAG]-T2A-hMC4R[3NG]] cells were plated onto 15 cm plates at a density of 3 million cells/plate and allowed to grow for 24 h. Cells were starved in medium devoid of serum to promote ciliation, and 18 h after the medium change, the cells were washed with ice-cold 1× PBS and were scraped off on ice in 1 ml of urea lysis buffer (10 mM HEPES, 130 mM NaCl, 3.6 mM KCl, 2.5 mM $MgCl_2$, 1.2 mM $CaCl_2$, 0.02 mM EDTA, 5 mM N-ethylmaleimide, 1 mM PMSF, 4 M urea, 0.25% w/v n-dodecyl β-D-maltoside). Equal amounts of IMCD3 and IMCD3-[MRAP2/MC4R] lysates were incubated with UBD resin pre-equilibrated in column buffer (50 mM Tris-HCl, 150 mM NaCl, 1 mM EDTA, 0.25% w/v n-dodecyl β-D-maltoside, 10% glycerol (pH 7.5)). The resins were washed by passing sequentially through 15 volumes of column buffer with 4M Urea, Wash Buffer I (50 mM Tris-HCl, 150 mM NaCl, 0.05% Tween-20 (pH 7.5)) and 15 volumes of Wash Buffer II (50 mM Tris-HCl, 1 M NaCl (pH 7.5)). The bound proteins were eluted by incubating the resin with 2 to 3 resin volumes of 1× SDS/LDS sample buffer in presence of DTT.

## Molecular modeling

Complex structural models of human MC4R bound to MRAP2 were generated with AlphaFold2 (version 2.3) in multimer mode, as implemented on ColabFold1.5.5 [41–43]. To cross-check the AF2 results, the newly released AlphaFold3 algorithm was accessed via the DeepMind portal (https://alphafoldserver.com). Protein structures were viewed and manipulated with PyMOL2.5 (https://pymol.org).

## Supporting information

**S1 Fig. Comparisons of GPR161[3NG] vs. MC4R[3NG] ciliary levels, structural models of MRAP2, and effects of the putative MC4R blocking peptide of MRAP2 on ciliary localization of MC4R. (A)** Split values of the $n = 3$ independent experiments comparing the ciliary fluorescence intensity of GPR161[3NG] vs. MC4R[3NG] in **Fig 1B**. Data points belonging to each

different experiment are encoded by translucent points of different color and shape. The average of each experiment is represented by solid points and the error bars represent 95% confidence interval. All underlying data are found S1 Data. (**B**) AlphaFold2.3 model of MRAP2 dimer. (**C**) AlphaFold2.3 model of human MC4R/MRAP2 in a 1:1:1 complex with human α-MSH. (**D**) Split values of the $n$ = 3 independent experiments in **Fig 4E** comparing the MC4R$^{3NG}$ ciliary fluorescence intensity in IMCD3-[MC4R$^{3NG}$] untransfected or transiently transfected cells with MRAP2$^{3FLAG}$ or the MRAP2 version lacking the candidate MC4R inhibitory motif MRAP2$^{\Delta block-3FLAG}$. Data points belonging to each different experiment are encoded by translucent points of different color and shape. The average of each experiment is represented by solid points and the error bars represent 95% confidence interval. All underlying data are found S1 Data.
(TIF)

**S2 Fig. Effects of co-treatment with AgRP and α-MSH on ciliary levels of MC4R and MRAP2.** (**A**) Split values of the $n$ = 3 independent experiments in **Fig 3B** comparing the MC4R$^{3NG}$ ciliary fluorescence intensity in IMCD3-[MRAP2$^{3FLAG}$;MC4R$^{3NG}$] cells treated with either vehicle or 50 nM AgRP ± α-MSH at different concentrations. Data points belonging to each different experiment are encoded by translucent points of different color and shape. The average of each experiment is represented by solid points and the error bars represent 95% confidence interval. All underlying data are found S1 Data. (**B**) Split values of the $n$ = 3 independent experiments in **Fig 3C** comparing the MRAP2$^{3FLAG}$ ciliary fluorescence intensity in IMCD3-[MRAP2$^{3FLAG}$;MC4R$^{3NG}$] cells treated with either vehicle or 50 nM AgRP ± α-MSH at different concentrations. Data points belonging to each different experiment are encoded by translucent points of different color and shape. The average of each experiment is represented by solid points and the error bars represent 95% confidence interval. All underlying data are found S1 Data.
(TIF)

**S3 Fig. Entry of MC4R into cilia is very inefficient in the absence of MRAP2.** (**A**) Split values of the $n$ = 3 independent experiments in **Fig 5B** comparing the MC4R$^{3NG}$ ciliary fluorescence intensity in cells expressing MC4R$^{3NG}$ alone or co-expressing MRAP2$^{3FLAG}$/MC4R$^{3NG}$ and treated with HS014, AgRP, or vehicle. Data points belonging to each different experiment are encoded by translucent points of different color and shape. The average of each experiment is represented by solid points and the error bars represent 95% confidence interval. All underlying data are found S1 Data. (**B**) Western blot of all $n$ = 3 individual experiments showing proof of partial Arl6 knock-down after treatment with a siRNA targeted to Arl6 in both cells expressing MC4R$^{3NG}$ alone or co-expressing MRAP2$^{3FLAG}$/MC4R$^{3NG}$. siRNA against luciferase (siLuc2) was used as a negative control. (**C**) Split values of the $n$ = 3 independent experiments in **Fig 5D** comparing the MC4R$^{3NG}$ ciliary fluorescence intensity in cells knocked-down for Arl6. Data points belonging to each different experiment are encoded by translucent points of different color and shape. The average of each experiment is represented by solid points and the error bars represent 95% confidence interval. All underlying data are found S1 Data. (**D**) Split values of the $n$ = 3 independent experiments in **Fig 5F** comparing the MC4R$^{3NG}$ ciliary fluorescence intensity in cells transiently transfected with $^{V5}$MRAP2. Data points belonging to each different experiment are encoded by translucent points of different color and shape. The average of each experiment is represented by solid points and the error bars represent 95% confidence interval. All underlying data are found S1 Data.
(TIF)

**S4 Fig. Co-trafficking of MC4R and MRAP2 into and out of cilia.** (**A**) Representative images of transiently transfected IMCD3 cells with either MRAP1$^{3FLAG}$ or MRAP2$^{3FLAG}$ and/or MC4R$^{GFP}$. MC4R$^{GFP}$ was visualized through the intrinsic fluorescence of eGFP (yellow). Serum-starved cells were fixed and stained for acetylated tubulin (acTub, magenta), FLAG (MRAP1/2$^{3FLAG}$, white), and DNA (cyan). The white, yellow, and magenta channels are shifted to facilitate visualization of ciliary signals in the insets. Scale bars: 5 μm (main panel) and 1 μm (inset). (**B**) Plots of ciliary MRAP1$^{3FLAG}$ or MRAP2$^{3FLAG}$ transiently transfected alone or in combination with MC4R$^{GFP}$ in IMCD3 cells ($n = 1$). While MRAP1 does not localize to cilium, MRAP2 can localize to the primary cilium on its own. Co-expression of MC4R increases ciliary enrichment of MRAP2. The solid line represents the median; the dashed lines are used to represent the interquartile values. Asterisks indicate statistical significance values calculated by a one-way ANOVA on individual cilia followed by a Holm–Sidak post hoc test (* $p < 0.05$, **** $p < 0.0001$). All underlying data are found S1 Data. (**C**) Representative images of transiently transfected IMCD3 cells with MRAP2$^{3FLAG}$ and subsequently treated with AgRP, HS014, or vehicle upon serum starvation for 24 h. Cells were fixed and stained for acetylated tubulin (acTub, magenta), FLAG (MRAP1/2$^{3FLAG}$, white), and DNA (cyan). The white and magenta channels are shifted to facilitate visualization of ciliary signals in the insets. Scale bars: 5 μm (main panel) and 1 μm (inset). (**D**) Superplot comparing the ciliary fluorescence intensity of MRAP2$^{3FLAG}$ in transiently transfected IMCD3 cells and treated with AgRP, HS014, or vehicle. $n = 3$ independent experiments. Data points belonging to each different experiment are encoded by translucent points of different color and shape. The average of each experiment is represented by solid points. A solid line has been used to represent the median of global data and dashed lines to represent the interquartile values. Statistical significance calculated by one-way ANOVA on individual cilia (n.s., nonsignificant, $p = 0.9367$). All underlying data are found S1 Data.
(TIF)

**S5 Fig. Constitutive MC4R removal from cilia depends on β-arrestin and the BBSome.** (**A**) Split values of the $n = 3$ independent experiments in **Fig 6B** comparing the MC4R$^{3NG}$ ciliary fluorescence intensity in cells co-expressing MRAP2$^{3FLAG}$/MC4R$^{3NG}$ in wild-type, Arl6$^{-/-}$ and Barr1$^{-/-}$/Barr2$^{-/-}$ genetic backgrounds. Data points belonging to each different experiment are encoded by translucent points of different color and shape. The average of each experiment is represented by solid points and the error bars represent 95% confidence interval. All underlying data are found S1 Data. (**B**) Split values of the $n = 3$ independent experiments in **Fig 6C** comparing the MRAP2$^{3FLAG}$ ciliary fluorescence intensity in cells co-expressing MRAP2$^{3FLAG}$/MC4R$^{3NG}$ in wild-type, Arl6$^{-/-}$ and Barr1$^{-/-}$/Barr2$^{-/-}$ backgrounds. Data points belonging to each different experiment are encoded by translucent points of different color and shape. The average of each experiment is represented by solid points and the error bars represent 95% confidence interval. All underlying data are found S1 Data.
(TIF)

**S6 Fig. Acceptor lysines required for ciliary exit are distributed on MC4R and its ciliary partners.** (**A**) Split values of the $n = 3$ independent experiments in **Fig 8B** comparing the ciliary fluorescence intensity of MC4R$^{3NG}$ in cells co-expressing MRAP2$^{3FLAG}$/MC4R$^{3NG}$, transiently transfected with cilia-targeted K63-specific deubiquitinase catalytic domain of AMSH (cilia-K63DUB), the catalytically inactive counterpart (cilia-K63DUB$^{†}$), or the cilia targeting sequence alone fused to the mScarlet reporter (cilia-reporter, CTS). Data points belonging to each different experiment are encoded by translucent points of different color and shape. The average of each experiment is represented by solid points and the error bars represent 95% confidence interval. All underlying data are found S1 Data. (**B**) Split values of the $n = 3$

independent experiments in **Fig 8E** comparing the ciliary fluorescence intensity of MC4R$^{3NG}$ in non-clonal IMCD3 lines stably co-expressing wild-type MRAP2$^{3FLAG}$/MC4R$^{3NG}$ (wt superscript) or the ubiquitination-refractory variants where the intracellular lysines have been substituted by arginine residues (K0 superscript). Data points belonging to each different experiment are encoded by translucent points of different color and shape. The average of each experiment is represented by solid points and the error bars represent 95% confidence interval. All underlying data are found S1 Data. (**C**) Superplot comparing the ciliary fluorescence intensity of MRAP2$^{3FLAG}$ in non-clonal IMCD3 lines stably co-expressing wild-type MRAP2$^{3FLAG}$/MC4R$^{3NG}$ (wt superscript, serpentine schematic without empty circles), or the ubiquitination-refractory variants where the intracellular lysines have been substituted by arginine residues (K0 superscript, serpentine schematic with empty circles indicating the location of the lysine-to-arginine substitution). $n = 3$ independent experiments. Data points belonging to each different experiment are encoded by translucent points of different color and shape. The average of each experiment is represented by solid points. A solid line has been used to represent the median of global data and dashed lines to represent the interquartile values. Asterisks indicate statistical significance values calculated by one-way ANOVA on individual cilia followed by Tukey post hoc test (*** $p < 0.001$; **** $p < 0.0001$). All underlying data are found S1 Data. (**D**) Split values of the $n = 3$ independent experiments in the previous panel comparing the ciliary fluorescence intensity of MRAP2$^{3FLAG}$ in non-clonal IMCD3 lines stably co-expressing wild-type MRAP2$^{3FLAG}$/MC4R$^{3NG}$ (wt superscript) or the ubiquitination-refractory variants where the intracellular lysines have been substituted by arginine residues (K0 superscript). Data points belonging to each different experiment are encoded by translucent points of different color and shape. The average of each experiment is represented by solid points and the error bars represent 95% confidence interval. All underlying data are found S1 Data. (**E**) Split values of the $n = 3$ independent experiments in **Fig 8G** comparing the ciliary fluorescence intensity of MC4R$_{cK0}$$^{3NG}$ in cells co-expressing MRAP2$_{cK0}$$^{3FLAG}$/MC4R$_{cK0}$$^{3NG}$, transiently transfected with cilia-targeted K63-specific deubiquitinase catalytic domain of AMSH (cilia-K63DUB), the catalytically inactive counterpart (cilia-K63DUB$^{†}$) or the cilia targeting sequence alone fused to the mScarlet reporter (cilia-reporter, CTS). Data points belonging to each different experiment are encoded by translucent points of different color and shape. The average of each experiment is represented by solid points and the error bars represent 95% confidence interval. All underlying data are found S1 Data.
(TIF)

**S1 Table. Primary antibodies and dilutions.** List of antibodies used in immunoblotting and immunofluorescence.
(XLSX)

**S1 Data. Supporting data for figures.** Excel sheets provide data, grouped in respective tabs, for Figs 1B and S1A; 1C; 2D; 2E; 3B and S2A; 3C and S2B; 4E and S1D; 5B and S3A; 5D and S3C; 5F and S3D; 6B and S5A; 6C and S5B; 8B and S6A; 8E and S6B; 8F and S6E; S4B; S4D; and S6C.
(XLSX)

**S1 Raw Images. Uncropped immunoblot images.** The uncropped immunoblots for Figs 2B, 2C, 4F, 6D, 7A, 7B, and S3B are shown.
(PDF)

# Acknowledgments

We thank Mark Hochstrasser for the gift of pET21-Cys-6xHis-otUBD plasmid and help with UBD-based purifications, David Komander for the gift of pOPINB-AMSH plasmid, Rohan

Baker for the gift of pET15b-USP2cc plasmid, Yien-Ming Kuo for help with microscopy, the HDFCCC Laboratory for Cell Analysis core personnel, Douglas Gould and his lab members for assistance with qRT-PCR experiments, Mark von Zastrow and Jeremy Reiter for comments on the manuscript, and all members of the Nachury lab for stimulating discussions.

## Author Contributions

**Conceptualization:** Irene Ojeda-Naharros, J. Fernando Bazan, Christian Vaisse, Maxence V. Nachury.

**Data curation:** Irene Ojeda-Naharros.

**Formal analysis:** Irene Ojeda-Naharros.

**Funding acquisition:** Irene Ojeda-Naharros, Christian Vaisse, Maxence V. Nachury.

**Investigation:** Irene Ojeda-Naharros, Tirthasree Das, Maxence V. Nachury.

**Methodology:** Irene Ojeda-Naharros, J. Fernando Bazan.

**Project administration:** Maxence V. Nachury.

**Resources:** Ralph A. Castro.

**Supervision:** Maxence V. Nachury.

**Validation:** Irene Ojeda-Naharros.

**Visualization:** Irene Ojeda-Naharros, J. Fernando Bazan.

**Writing – original draft:** Irene Ojeda-Naharros, Maxence V. Nachury.

**Writing – review & editing:** Irene Ojeda-Naharros, Tirthasree Das, Ralph A. Castro, J. Fernando Bazan, Christian Vaisse, Maxence V. Nachury.

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
