## [Editor Report · Decision Letter 0]

16 Sep 2024

Dear Dr Nachury, 

Thank you for submitting your manuscript entitled "Tonic ubiquitination of MC4R promotes its constitutive exit from cilia" for consideration as a Research Article by PLOS Biology.

Your manuscript has now been evaluated by the PLOS Biology editorial staff as well as by an academic editor with relevant expertise and I am writing to let you know that we would like to send your submission out for external peer review.

Once your full submission is complete, your paper will undergo a series of checks in preparation for peer review. After your manuscript has passed the checks it will be sent out for review. To provide the metadata for your submission, please Login to Editorial Manager (https://www.editorialmanager.com/pbiology) within two working days, i.e. by Sep 18 2024 11:59PM.

Kind regards,

Ines

--

Ines Alvarez-Garcia, PhD

Senior Editor

PLOS Biology

---

## [Decision Letter · Decision Letter 1]

13 Oct 2024

Dear Dr Nachury,

Thank you for your patience while your manuscript entitled "Tonic ubiquitination of MC4R promotes its constitutive exit from cilia" was peer-reviewed at PLOS Biology. It has now been evaluated by the PLOS Biology editors, an Academic Editor with relevant expertise, and by three independent reviewers. 

The reviews are attached below. As you will see, the reviewers find the conclusions novel and significant for the field, but they also raise several issues that should be addressed before we can consider the manuscript for publication. Reviewer 1 asks for several clarifications of the data and also to show protein levels of GPR161 and MC4R to support the claim that MC4R is expressed at higher levels, or to provide transcript levels if the tonic activity-mediated degradation complicates the analysis. Reviewer 2 thinks that western blots or RT-QPCR data for the cells shown on Fig. 1 and S1 should be added to allow comparison of total cellular expression levels of the MC4R and GPR161 transgenes. In addition, this reviewer suggests that testing how AgRP or HS014 treatment affects ciliary levels of MRAP2 alone would complete the study, along with some statistics and replicates that are missing in Fig. S5. Reviewer 3 is very positive and only has a couple of minor comments to improve the discussion.

In light of the reviews, we would like to invite you to revise the work to thoroughly address the reviewers' reports. Please note that we cannot make a decision about publication until we have seen the revised manuscript and your response to the reviewers' comments. Your revised manuscript is likely to be sent for further evaluation by all or a subset of the reviewers.

**IMPORTANT - SUBMITTING YOUR REVISION**

3. Resubmission Checklist

a) *PLOS Data Policy*

b) *Published Peer Review*

d) *Blurb*

Please also provide a blurb which (if accepted) will be included in our weekly and monthly Electronic Table of Contents, sent out to readers of PLOS Biology, and may be used to promote your article in social media. The blurb should be about 30-40 words long and is subject to editorial changes. It should, without exaggeration, entice people to read your manuscript. It should not be redundant with the title and should not contain acronyms or abbreviations. For examples, view our author guidelines: https://journals.plos.org/plosbiology/s/revising-your-manuscript#loc-blurb

Sincerely,

Ines

--

Ines Alvarez-Garcia, PhD

Senior Editor

PLOS Biology

Reviewers' comments

Rev. 1:

In this manuscript, the authors describe the mechanism mediating the constitutive exiting of MC4R from the primary cilium. The study is rigorous and novel. The data are compelling and support the proposed model.

Minor comments:

1. The authors show that adding the MC4R antagonist AgRP results in MC4R ciliary accumulation. Is this reversed with co-treatment of AgRP and a-MSH to re-activate MC4R?

2. The authors state that the fact that the accumulation of MC4R with AMSH (Figure 7B) is greater than with the MC4R-cKO (Figure 7E) suggests the presence of additional Ub acceptors. However, Figure 7E uses a non-clonal cell line with much higher basal ciliary MC4R expression than the isogenic cell line used in 7B, confounding this interpretation. Please also comment on why MRAP2-cKO is not sufficient to prevent MC4R exit.

3. Please show protein levels of GPR161 and MC4R (Figure 1) to support the claim that MC4R is expressed at higher levels. Alternatively, if the tonic activity-mediated degradation complicates this analysis, provide transcript levels.

4. In the methods section, include the methodology for calculating the number of MC4R receptors in cilia.

5. In the discussion section, the paragraph discussing the role of GRK2 is too speculative given the data presented.

6. Figure 3B - inset is in the wrong location

Keren Hilgendorf

Rev. 2:

Disclaimer: I am not an expert in structural biology and cannot give a qualified assessment of the data shown in Figure 3A, C and Figure S1B, C.

In this study, the authors address the molecular mechanisms that regulate the ciliary localization and homeostasis of the GPCR melanocortin receptor 4 (MC4R), which is an essential regulator of body weight homeostasis. Previous work by the authors and others showed that MC4R needs to be targeted to primary cilia in hypothalamic neurons to regulate feeding behavior, yet MCR4 is almost undetectable in primary cilia in the adult brain under ad lib fed conditions. To address this apparent paradox, the authors use a cell-based model system (IMCD3 cells) combined with AlphaFold structural modeling to dissect the molecular mechanisms that regulate the ciliary localization and homeostasis of MC4R. They convincingly show that constitutively active MC4R continuously exits cilia by a mechanism depending on beta-arrestin, ubiquitination and the BBSome, a ciliary trafficking complex previously shown to promote export of ubiquitinated GPCRs from cilia. Conversely, blocking MC4R activity by the inverse agonists AgRP or HS014 causes it to accumulate in cilia. Using cells expressing tagged versions of both MC4R and its partner MRAP2, the authors confirm previous observations indicating that MRAP2 is required for ciliary entrance of MC4R, whereas MRAP2 does not seem to affect the tonic activity or stability of MCR4.

Overall, I find that the paper is timely and well written, and the data quality is high. One could question the use of the IMCD3 cell model as opposed to a neuronal cell type, but on the other hand, the IMCD3 cell model used by the authors seems to be a "clean" system offering experimental control over relevant proteins and ligands. I have a few relatively minor comments/suggestions that the authors should address prior to publication of this work.

Main comments

1) Figure 1, Figure S1 and 1st paragraph of results: western blots or RT-qPCR data should be provided for these cells to allow comparison of total cellular expression levels of the MC4R and GPR161 transgenes.

2) Figure 2D, E: I am puzzled as to why the ciliary MRAP2-3FLAG levels increase so dramatically upon AgRP or HS014 treatment if ciliary entrance of MRAP2 alone does not rely on MC4R (Figure S5A-B; Bernhard et al., 2023)? Please comment. To substantiate their model, it would be really nice if the authors could test how AgRP or HS014 treatment affects ciliary levels of MRAP2 alone, i.e. in the absence of MC4R.

3) In Figure S5, statistical analysis is missing and there is no indication about number of biological replicates. Also, are the total cellular expression levels of MRAP1-FLAG and MRAP2-FLAG identical in this setup?

Very minor comment

1) Introduction, 9 lines from bottom: the word "syndromic" before "obesity syndrome" seems redundant and should be removed.

Rev. 3:

The causal link between ciliary dysfunction and obesity is well established in human and animal models. GPCR MC4R is an important regulator of body weight homeostasis, and its ciliary localization has been shown (by the same group) to be key in this process. In this manuscript, Ojeda-Naharros and colleagues provided careful characterizations and convincing evidence that detailed the ciliary exit mechanisms of MC4R. Overall, the design of the study is elegant and thorough, and the quality of data is high. This reviewer only has a couple of minor comments:

1. "SMO offers one exception…" paragraph in the discussion provides interesting insights regarding ciliary GPCR trafficking comparisons, but seems out of place since this paper is only focused on MC4R. It would be better to integrate the Smo information more tightly into the discussion about MC4R or to simply cut the paragraph.

2. The importance of ciliary localization of MC4R in the prevention of obesity is hard to understand and is a key question to answer regarding the significance of MCR4 in obesity and ciliopathies. Although this question may be out of the scope of this study, a more thorough discussion (rather than only a few sentences in the end of the discussion) would be appreciated. The comment that "MC4R may reach its relevant physiological target immediately after exiting cilia, rather than inside cilia" seems interesting but difficult to understand because not much background information is provided. If MC4R has a strong tonic activity outside of cilia, why wouldn't the non-ciliary MC4R reach their targets, if the targets, as the author hinted, are localized near the base of cilia or the surrounding area in the cytoplasm?

---

## [Editor Report · Decision Letter 2]

18 Dec 2024

Dear Dr Nachury,

Thank you for your patience while we considered your revised manuscript entitled "Tonic ubiquitination of MC4R promotes its constitutive exit from cilia" for publication as a Research Article at PLOS Biology. This revised version of your manuscript has been evaluated by the PLOS Biology editors and by the Academic Editor.

Based on our Academic Editor's assessment of your revision, we are likely to accept this manuscript for publication, provided you satisfactorily address the data and other policy-related requests stated below.

In addition, we would like you to consider a suggestion to improve the title:

"Tonic ubiquitination of the essential body weight regulator melanocortin receptor 4 promotes its constitutive exit from cilia in hypothalamic neurons"

We expect to receive your revised manuscript by January 6, but do let us know if you need more time given the holidays. 

*Published Peer Review History*

*Press*

Sincerely,

Ines

--

Ines Alvarez-Garcia, PhD

Senior Editor

PLOS Biology

DATA POLICY:

Many thanks for providing the data underlying the graphs shown in the figures. I have checked the file you sent us and we are missing some data, thus please provide a new file containing also the data underlying the graphs shown in the following figures:

Fig. S1A, D; Fig. S2A, B; Fig. S3A, C, D; Fig. S5A, B and Fig. S6A, B, D, E

In addition, please ensure that all the corresponding figure legends in your manuscript include information on where the underlying data can be found, including the supplementary ones.

CODE POLICY

---

## [Editor Report · Decision Letter 3]

17 Jan 2025

Dear Dr Nachury,

Thank you for the submission of your revised Research Article entitled "Tonic ubiquitination of the central body weight regulator melanocortin receptor 4 (MC4R) promotes its constitutive exit from cilia" for publication in PLOS Biology. On behalf of my colleagues and the Academic Editor, Dagmar Wachten, I am delighted to let you know that we can in principle accept your manuscript for publication, provided you address any remaining formatting and reporting issues. These will be detailed in an email you should receive within 2-3 business days from our colleagues in the journal operations team; no action is required from you until then. Please note that we will not be able to formally accept your manuscript and schedule it for publication until you have completed any requested changes.

PRESS

Sincerely, 

Ines

--

Ines Alvarez-Garcia, PhD

Senior Editor

PLOS Biology
